# Exploring Health Outcomes for U.S. Veterans Compared to Non-Veterans from 2003 to 2019

**DOI:** 10.3390/healthcare9050604

**Published:** 2021-05-18

**Authors:** Jose A. Betancourt, Paula Stigler Granados, Gerardo J. Pacheco, Julie Reagan, Ramalingam Shanmugam, Joseph B. Topinka, Bradley M. Beauvais, Zo H. Ramamonjiarivelo, Lawrence V. Fulton

**Affiliations:** School of Health Administration, Texas State University, San Marcos, TX 78666-4684, USA; psgranados@txstate.edu (P.S.G.); gjp46@txstate.edu (G.J.P.); jkr73@txstate.edu (J.R.); shanmugam@txstate.edu (R.S.); josephtopinka@txstate.edu (J.B.T.); bmb230@txstate.edu (B.M.B.); zhr3@txstate.edu (Z.H.R.); lf25@txstate.edu (L.V.F.)

**Keywords:** U.S. Veteran, deployment, spatial regression, obesity, comorbidities, risk-factors, overweight, diabetes

## Abstract

The physical demands on U.S. service members have increased significantly over the past several decades as the number of military operations requiring overseas deployment have expanded in frequency, duration, and intensity. These elevated demands from military operations placed upon a small subset of the population may be resulting in a group of individuals more at-risk for a variety of debilitating health conditions. To better understand how the U.S Veterans health outcomes compared to non-Veterans, this study utilized the U.S. Centers for Disease Control and Prevention (CDC) Behavioral Risk Factor Surveillance System (BRFSS) dataset to examine 10 different self-reported morbidities. Yearly age-adjusted, population estimates from 2003 to 2019 were used for Veteran vs. non-Veteran. Complex weights were used to evaluate the panel series for each morbidity overweight/obesity, heart disease, stroke, skin cancer, cancer, COPD, arthritis, mental health, kidney disease, and diabetes. General linear models (GLM’s) were created using 2019 data only to investigate any possible explanatory variables associated with these morbidities. The time series analysis showed that Veterans have disproportionately higher self-reported rates of each morbidity with the exception of mental health issues and heart disease. The GLM showed that when taking into account all the variables, Veterans disproportionately self-reported a higher amount of every morbidity with the exception of mental health. These data present an overall poor state of the health of the average U.S. Veteran. Our study findings suggest that when taken as a whole, these morbidities among Veterans could prompt the U.S. Department of Veteran Affairs (VA) to help develop more effective health interventions aimed at improving the overall health of the Veterans.

## 1. Introduction

The collective health of those who served in the U.S. military exhibits a number of health conditions that paint a concerning picture of morbidity among the U.S. Veteran population [1,2,3,4,5]. Veterans are not only found to suffer from obesity at higher levels when compared to the average U.S. population [6,7], but also suffer from a significantly higher level of associated co-morbidities including coronary heart disease-CHD or angina; stroke; skin cancer; other cancers; chronic obstructive pulmonary disease (COPD), emphysema, or chronic bronchitis; arthritis, rheumatoid arthritis, gout, lupus, or fibromyalgia; depressive disorders; and diabetes [1,8,9,10]. Particularly concerning is the interaction of depression, diabetes, and obesity and their associated health outcomes [6,11,12]. Coupled with the growing ‘obesity epidemic’ impacting the U.S., it is important to identify geographic locations where Veteran populations reside so as to better resource health intervention programs aiming to meet the health needs of Veterans [13,14,15]. By analyzing population-based, health data for Veterans when compared to the U.S. population, this may better identify the population of U.S. military Veterans (by specific demographic characteristics) who could benefit from health systems interventions aimed at solutions for reducing weight and reducing morbidity from certain health risks [1,15].

### 1.1. Stressors Associated with Military Service and Deployments

Military service is associated with a number of physical and psychological stressors including “deploying to overseas missions, participating in combat operations, sustaining trauma and/or serious physical and/or mental injuries, and ultimately reintegrating back into civilian life” [6,16]. These stressors may have immediate negative effects such as physical or psychological injury and/or trauma, or delayed effects that do not manifest until later in life [16]. Additional stressors do not necessarily involve deployments nor combat, but simply maintaining requisite levels of physical fitness and height/weight standards in accordance with military service standards [17]. Every Active-duty Service member (ADSM) is required to meet strict height and weight standards which are evaluated every six months in order to remain employed. This is normally not a challenge to Service members who regularly exercise with their units and maintain healthy eating habits. However, those individuals who do not exercise nor practice good eating habits on a regular basis may find themselves close to exceeding their weight limits at the time of their fitness test. In order to “make weight” or meet weight standards, a number of Service members resort to unhealthy behaviors including excessive exercise, fasting/skipping meals, using sauna/rubber suits, laxatives, diuretics, and in extreme cases, inducing vomiting [17]. Researchers have found that these unhealthy behaviors often result in maladaptive eating behaviors later in life including binge eating, vomiting, emotional eating, food addiction, and night eating: all contributors to obesity [17].

### 1.2. Operational Tempo (OPTEMPO)/Personnel Tempo (PERSTEMPO)

Each year, the U.S. Congress determines the number of personnel assigned to the U.S. Department of Defense (DoD) as required by law. Figure 1 depicts the number of Active-duty personnel by Service (Army, Navy, Marine Corps, and Air Force) from 1950 through the year 2000 [18]. Figure 1 illustrates sharp increases in total personnel strength levels in times of conflict such as during the Korean War and Vietnam Conflict. More personnel available for deployment normally allows for the average Service member to be deployed only one time during their total military experience. However, with the advent of the all-volunteer force in the early 1970s [19], the total number of military personnel available for deployment began a steady decline in total size, which as depicted in Figure 1 continues to the present day.

Unfortunately, the pace of military operations (Operational Tempo or OPTEMPO) has not aligned well with this reduction in the size of the force. Figure 2 not only depicts the major military conflicts from 1940 (World War II) through the present, but also the duration of each conflict, which when compared to the previous figure (Figure 1) may suggest that with a reduced pool of deployable personnel, the average Service member will likely deploy MORE than one time during their military experience [20].

Figure 3 further specifically focuses on the period 1980 through the present (2020) and illustrates a sharp increase in the number of military operations at a time when there have been a smaller number of available Service members, resulting in an increased personnel tempo (PERSTEMPO).

As depicted in Figure 2 and Figure 3, the number, length, and duration of military operations since the mid to late 1980s through the present has intensified, while the size of the U.S. military has gradually declined throughout this period. This combination of increased demand in military operations placed on a reduced supply of available and deployable personnel has resulted in a military force that is exhibiting the results of these stressors in poor health outcomes [1]. The U.S. DoD has attempted to increase the available pool of personnel for military deployments by utilizing additional military units from the Reserve Component (RC) as well as the National Guard [20]. However, some prior research indicates this only increased the number of Veterans with poor health outcomes [1]. The specific aim of this study is to provide a snapshot or a “glimpse” into potential associations between the stressors associated with multiple deployments, and poor health outcomes among US Veterans. We specifically investigate morbidities between Veteran and non-Veterans and over time with the belief that Veterans are likely to experience more morbidities due to the difficulties and exposures of military service. Further, we use data from 2003 to 2010 and aggregated data from 2011 to 2019, which include the newest release of the Centers for Disease Control and Prevention (CDC) Behavioral Risk Factor Surveillance System (BRFSS) [21,22,23] dataset, to look at demographic, socio-economic, geographic, time, and Veteran status variables that may assist in explaining any differences.

## 2. Materials and Methods

To compare Veteran and non-Veteran health outcomes, yearly age-adjusted, population estimates from 2003 (or earliest year available) to 2019 were used for Veteran/non-Veteran. Sampling weights were applied to evaluate the panel series for each of the separate morbidities (overweight/obesity, heart disease, stroke, skin cancer, cancer, COPD, arthritis, mental health issues, kidney disease, and diabetes). Confidence intervals were graphically constructed. General linear models (GLM’s) with quasi-binomial link functions, which account for fractional observations due to weighting, were then created by morbidity from the year 2011 forward. Because of the change in BRFSS sampling methods beginning in 2011, the CDC warns that “The BRFSS 2011 data should be considered a baseline year for data analysis and is not directly comparable to previous years of BRFSS data because of the changes in weighting methodology and the addition of the cell phone sampling frame [24].”

### 2.1. Data Source

The BRFSS is a continuous health survey system that collects information from each state in the U.S., three U.S. territories and the District of Columbia about modifiable risk factors for chronic diseases and other leading causes of death [25]. The BRFSS is conducted by state health departments annually with assistance from the CDC using in-house interviewers or contracts with telephone call centers or universities [25,26,27]. There is a standardized core questionnaire with optional modules, and some states add in their own questions. The survey uses Random Digit Dialing (RDD) techniques on both landlines and cell phones to conduct the interviews. An estimated 400,000 surveys are conducted annually [26]. A full description of the methodology can be found online [28]. The BRFSS is a publicly available, anonymous data set that does not require an institutional review board approval or an exempt status for its use.

### 2.2. Study Sample


For this study, BRFSS data from 2003 to 2019 (last available year) were acquired from the CDC [26]. Each year was used to estimate the population point estimate for the specific time and morbidity. The BRFSS data are considered by the CDC to be the nation’s best source for health-related survey data, and the 2019 version includes 418,268 observations. Applying stratum and individual weights to these observations results in a picture of the entire nation’s self-reported health [29]. In total, this study included 7.181 million sample observations.

Table 1 shows the population estimates and sample sizes for each of the years in question by Veteran/Non-Veteran status. Sampling weights based on location and socio-demographic information were applied to estimate the entire US population, resulting in fractional values for population estimates. Prior to 2011, samples were weighted by stratum (area code/prefix combinations) times the inverse number of telephones in the household times the number of adults in the household times the number of people in an age-by-sex or age-by-race/ethnicity-by-sex category. This was done in the population of a region or a state divided by the sum of the preceding weights for the respondents in the same age-by-sex or age-by-race/ethnicity-by-sex category (adjusting for noncoverage and nonresponse and forces the sum of the weighted frequencies to equal population estimates for the region or state.) There were two separate variable weight fields. This weighting was applied through the use of separate fields for stratum and weights. From 2011 onwards, raking replaced the post-stratification process. Raking allows for incorporation of cellular telephone survey data and allows for inclusion of other demographic considerations to improve socio-demographic matching by stratum.

### 2.3. Study Measures

Table 2 provides the variable names, question, and coding for dependent variables, controls, the primary independent variable, and the weighting variables. Over the course of two decades, several variable name and measurement changes occurred. Most were either nominal (e.g., Veteran status) or easily re-codable. A complete discussion of these recoded variables is available online [30]. One additional variable was included for longitudinal models. This variable, “YEAR” was an indicator for the year of the BRFSS.

#### 2.3.1. Dependent Variables

All dependent variables in the study were dichotomously coded. In all cases, the categories of “Don’t Know/Not Sure” and “Not Asked/Missing” were imputed with the modal response, as the proportion of these values was 1% or less. For obesity, the modal response was greater than 25% body mass index. For all other dependent variables, the modal response was “No.” For all variables other than obesity, the proportion missing was 1% or less (negligible). For obesity, about 10% of the observations were in the categories “Don’t Know/Refused/Missing.” The modal response reflecting overweight/obesity status (greater than 60% of the respondents) is likely to best categorize these individuals. Table 3 shows the unweighted and weighted proportions in the categories of “Yes”, “No”, and “Unknown” for the dependent variables in year 2019.

The mental health variable was unique in that there was not a specific variable that addressed the presence or absence of the disease. Instead, the question in the BRFSS survey was, “Now thinking about your mental health, which includes stress, depression, and problems with emotions, for how many days during the past 30 days was your mental health not good?” This is the MENTHLTH variable in the BRFSS dataset. To address this issue, we dichotomously coded mental health as “1” if there were any days in the past 30 days that an individual reported his or her mental health status as “not good.” Many of the dependent variables did not have observations dating back to 2003 (or had inconsistent add-on modules not universally asked of respondents). Table 4 shows the analysis starting year for each of the variables.

#### 2.3.2. Independent Variable

The primary variables of interest were the year of the estimate (2003 through 2019), and Veteran status, recoded {0 = did not self-report as Veteran, 1 = self-reported as veteran}. This coding results from the survey questions that follow.

Years 2003 through 2006 (Code: VETERAN). Have you ever served on active duty in the United States Armed Forces, either in the regular military or in a National Guard or military reserve unit?” 1 = Yes, 2 = No, 7 = Don’t Know/Not Sure, 9 = Refused, Blank = Not asked/Missing.Years 2007 through 2008 (Code: VETERAN1). “Have you ever served on active duty in the United States Armed Forces, either in the regular military or in a National Guard or military reserve unit? Active duty does not include training for the Reserves or National Guard, but DOES include activation, for example, for the Persian Gulf War.” 1 = Yes, 2 = No, 7 = Don’t Know/Not Sure, 9 = Refused, Blank = Not asked/Missing.Years 2009 (Code VETERAN2). “Have you ever served on active duty in the United States Armed Forces, either in the regular military or in a National Guard or military reserve unit? Active duty does not include training for the Reserves or National Guard, but DOES include activation, for example, for the Persian Gulf War.” 1 = Yes, now on Active Duty, 2 = Yes, on Active Duty during the last 12 months but not now, 3 = Yes, on active duty in the past, but not during the last 12 months, 4 = No, training for Reserves or National Guard only, 5 = No, never served in the military, 7 = Don’t Know/Not Sure, 9 = Refused, Blank = Not asked/Missing.Years 2010 through 2019 (Code: VETERAN3). “Have you ever served on active duty in the United States Armed Forces, either in the regular military or in a National Guard or military reserve unit? Active duty does not include training for the Reserves or National Guard, but DOES include activation, for example, for the Persian Gulf War.” 1 = Yes, 2 = No, 7 = Don’t Know/Not Sure, 9 = Refused, Blank = Not asked/Missing.

In all years, the responses were identically coded with the exception of 2009. Recoding for this year assigned any yes values equal to 1 and all other values equal to zero. Due to the exceedingly low numbers of responses that were blank, refused, and “do not know” (e.g., 0.574% in 2019), these were imputed with the modal response of non-Veteran (86.7% of the population in 2019).

#### 2.3.3. Covariates

##### Demographic Variables

Gender was dichotomously recoded (1 = self-identify as male, 0 = otherwise). In 2019, missing data were for this variable were imputed by BRFSS; in 2018, only 0.26% of all observations were don’t know/not sure or refused. The mode was imputed in these cases.

Race was coded as (1 = Caucasian, 0 = otherwise), and ethnicity was coded as (1 = Hispanic, 0 = otherwise). These variables were both generated from the *IMPRACE* or *RACE* variables, depending on the year. Completely discrete categories were readily attainable via recoding (see https://rpubs.com/R-Minator/BRFSS). Age was coded as (1 = age 18 to 24, 2 = age 25 to 36, 3 = age 35 to 44, 4 = age 45 to 54, 5 = age 55 to 64, 6 = age 65 or older), with the small proportion of missing (e.g., 1.6% in 2019) recoded to the modal category (age 65 or older). Marital status was also dichotomously coded (1 = married, 0 = otherwise). Again, the small proportion missing was negligible (e.g., 0.82% in 2019).

##### Socio-Economic Variables

Education status was a dichotomous variable, coded as (1 = self-identify as a college graduate, 0 = otherwise). Little data were missing from this variable (less than 0.5% in 2019). The variable income was also dichotomous with values (1 = self-reported modal income of $75,000 or more household income per year, 0 = otherwise). Employment was also evaluated as a dichotomous variable with (1 = employed for wages, the modal response, and 0 = otherwise).

##### Geographic Variable

The variable “region “was a feature engineered Census Bureau region based on the state/territory of residence based on the Federal Information Processing System (FIPS) Codes with (1 = New England, 2 = East North Central, 3 = East South Central, 4 = Middle Atlantic, 5 = Mountain, 6 = Pacific, 7 = South Atlantic, 8 = West North Central, 9 = West South Central, 10 = Territories). We collapsed the states into regions to provide better clarity of analysis. Definitions of the states in these regions are available here: https://www.census.gov/prod/1/gen/95statab/preface.pdf.

##### Time Variable

For the combined 2011 through 2019 datasets, a variable indicating the BRFSS year was created. This variable was treated as a factor-level variable to identify non-linear temporal effects.

### 2.4. Methods and Models

#### 2.4.1. Descriptive Models

Univariate age-adjusted models for Veterans and Non-Veterans were constructed for each dependent variable for each year to evaluate differences. These models were accompanied by sparkline graphs, which provide small multiples [31].

#### 2.4.2. General Linear Models

For GLM models using the 2011–2019 data, quasibinomial models were generated. Given the dichotomous nature of the dependent variables, a logistic regression (assuming binomial family link) would have been appropriate until weights were applied. When weighted, the dichotomous variables become fractional. The quasibinomial is then appropriate for these types of weighted survey analyses [32].

#### 2.4.3. Data Analysis

R Statistical Software [25,26,27] and R Studio [33] were used to conduct the analysis. The survey library in R [34] provided the weighting necessary to generate population estimates, and the EpiTools package [35] provided age-adjusted estimates to compare Veteran and non-Veteran populations.

## 3. Results

The entirety of the analysis is available for review online [29].

### 3.1. Descriptive Statistics

The age-adjusted comparisons for all dependent variables for all years by Veteran and non-Veteran status are shown in Table 5. For the years 2003 to 2010, unfortunately we were unable to combine these data due to differences in the weighting and data collection methods used by the CDC during this timeframe. Therefore, we examined these data independently and arrived at our best estimates. For this period, the results indicate that Veterans have higher rates for obesity, diabetes, and heart disease while stroke is relatively the same between the Veteran and non-Veteran populations. However, the Veteran population reported less mental health issues than the non-Veteran population. At the bottom of the table, the combined years for 2011 through 2019 are shown. The 2011–2019 data show that Veterans are generally more obese (72% vs. 60%) and have higher rates of diabetes (19% vs. 16%), heart disease (13% vs. 8%), stroke (6% vs. 5%), skin cancer (16% vs. 11%), cancer (13% vs. 11%), and COPD. The 2011–2019 estimated rate of arthritis (4%) is identical for both groups. Veterans have lower rates of mental health issues (30% vs. 37%) and arthritis (nominally, 37% vs. 38%), compared with the non-Veteran population. Trends from 2003 to 2019 for each variable are generally flat, as evidenced by the sparkline graphs and a review of the age-adjusted table values.

### 3.2. General Linear Models (GLMs)

The CDC suggests that surveys before 2011 should not be used as part of longitudinal analysis [36] due to changes in weighting. Thus, GLM models were built using only dependent variables from 2011 to 2019. A quasi-binomial link function was used, s, as it allowed fractional values of integers that occur when weights are applied [37]. These results are shown in Table 6. The GLM results depict odds ratios for all control variable groupings (demographics, socio-economics, geographical regions, and years) as well as for Veteran status and year for each of the dependent variables. Demographics and socio-economic variables account partially for socioeconomic determinants of health associated with disease processes. Geographical effects were considered as they may account for culture and other locality factors (environmental effects). Statistically significant odds ratios greater than 1 suggest a higher-than-average risk for all people within those categories (regardless of Veteran status). Veterans disproportionately self-report a statistically significant and higher amount of the 10 selected morbidities when compared to non-Veterans with only a single exception: mental health. The odds ratio for mental health is 0.972. These findings are congruent but not identical to the age-adjusted marginal analysis.

## 4. Discussion

Observing data over time, these results show that Veterans have disproportionately higher self-reported rates of stroke, skin cancer, cancer, COPD, arthritis, kidney disease, and diabetes. Similar to previous studies [1,2,3,4,5] these results appear to indicate Veteran populations are significantly less healthy than non-Veterans. Obesity was separated from the other variables as it was significantly higher for Veterans and may also contribute to the increased observance of the other morbidities [4].

Multiple deployments over the last several decades may be a major contributor to increased illness seen after separation from the service [38]. The demand placed upon the U.S. Service member, particularly since the mid to late 1980s has been illustrated in the number of military operations requiring deployment [20]. Unfortunately, the number of available personnel to support these deployments has not kept pace, resulting in more Service members deploying more often. As depicted in the health outcomes data, this combination of increased demand in military operations placed upon a smaller population may be resulting in a group of at-risk individuals for a variety of debilitating health conditions [1]. This study sought to utilize the BRFSS data to provide a snapshot or “glimpse” into a possible correlation between the stressors associated with multiple deployments, and poor health outcomes. Unfortunately, although the BRFSS dataset does illustrate those who identify themselves as ‘Veterans,’ it lacks the desired level of detail to fully define an association between multiple deployments and poor health outcomes. We hope that this study may prompt the Veterans Administration or similar responsible agencies to further refine the information collected on the Veteran population.

Additional stressors may exist in the occupational requirement for all Service members to meet stringent height and weight standards, and physical fitness standards, both of which are evaluated every six months [17]. Perhaps the combination of each of these stressors ultimately contribute to the stark differences in health outcomes between the Veteran populations compared to non-Veteran populations [1,2,3]. It is important to also note that self-reported mental health disorders and heart disease were not higher for Veterans when comparing to non-Veterans. It is possible that the negative consequences of reporting mental health issues in the military (e.g., security clearance revocation) continue to play a role in how persons reported in this dataset [39]. This phenomenon may also be a result of insufficient assessment for depression among the Veteran population by responsible agencies. Regardless, Veteran services to address the need for mental health services such as screening, assessment, and treatments continue to challenge the Veteran’s Administration (VA) and the population they serve. The need for mental health services for Veterans continues to spawn new, innovative services from the VA such as the use of “tele-mental health” capabilities [40]. In fact, the advent of the current, worldwide COVID-19 pandemic has served as a catalyst to promoting and utilizing such tele-mental health services by Veterans [40]. The lower reports of heart disease may be due in part to the exercise regimen that Veterans experience while in the military. Further, we note that the jump from 2003 to 2007 in heart disease proportion may be due to increased diagnoses by the clinical community [41]. Further research in the area is needed.

As shown in our results, demographics and socio-economic variables could account partially for socioeconomic determinants of health associated with disease processes.

As such some specific health screening, prevention, and health care services should be provided to specific population based on our findings. Regarding the demographic, socioeconomic and geographic variables, our findings suggest that odds of being overweight, being diagnosed with a heart disease, skin cancer, or other cancers, COPD, arthritis, kidney disease and diabetes, as well as the odds of having a stroke increase with age. Walker et al., 2019, also found that weight and age were positively associated with the odds of having a heart disease [42]. In addition, being a Caucasian is associated with increased odds of having a heart disease, skin cancer, other cancers, arthritis, COPD, and mental health issues, but associated with lower odds of being overweight, having a stroke, or a kidney disease compared with being a non-Caucasian. Hispanics have a better health status than their non-Hispanic counterparts. While Hispanics have higher odds of being overweight, having a skin cancer, and a kidney disease, they have lower odds of having a heart disease, stroke, other cancers, COPD, arthritis, or mental health issue compared with non-Hispanics. Our finding that Hispanics are less likely to have heart disease is supported by previous study on veterans [42].

Furthermore, compared to being a female, being a male is associated with higher odds of being overweight, as well as having a heart disease, stroke, skin cancer, and diabetes, but it is associated with lower odds of having other cancers, COPD, arthritis, mental health issues, and kidney disease. With regards to marital status and annual income, married individuals, and those with annual income of ≥ $75,000 have a better health status than singles and those who have lower income. They are less likely to have heart disease stroke, other cancers, COPD, arthritis, mental health issues, kidney disease, and diabetes, but more likely to be overweight and having skin cancer, compared with their single- and lower-income counterparts. In the same vein, being college graduates and employed for wages are associated with a better health status compared with those who are not collage graduate and are not employed for wages. Being a college graduate was associated with lower odds of all having these health issues except for skin cancer and being employed for wage was only associated with being overweight. Our findings regarding the associations between gender and heart disease as well as income and heart disease are supported by Walker et al., 2019 [42].

With respect to geographic location, individuals who live in the East North Central, Middle Atlantic, Mountain, Pacific, South Atlantic, West South Central regions, as well as in the Territories of U.S. are more likely to have skin cancer, while individuals who reside in the West North Central region are more likely to have a heart disease.”

Previous studies have also illustrated differences in burden of diseases, injuries, and risk factors in the United States by state [43]. It is interesting to note here that healthcare is provided to all Active-duty Service members while in the Service; therefore, access should not be an issue for the Service member. However, given this, there are still health disparities among the US military population primarily due to differences in health literacy, higher stress, and working conditions between those of lower rank and those of higher rank [44]. Further research must explore the factors of military rank, time in service, number of deployments, level of health literacy, and any additional factors which may account for differences in health outcomes in a population which has the same level of healthcare access. Another potential area of future study is how the length of time that a Veteran has been out of the Service impacts the Veteran’s overall health status. It may be plausible to conclude that being a Veteran shortly before the survey year compared to being a Veteran for several years before the survey year may also result in different health statuses. Additionally, geographical effects were considered as they may account for culture and other locality factors (environmental effects). We saw in our results how geographies and physical locations vary and result in different health outcomes. Again, more research is needed in this area to further inform agencies such as the VA as to where to target resources that can meet the health needs of Veteran populations suffering from poorer health.

The data for 2019 indicate that Veteran status is associated with significantly different odds for all morbidities except for mental health disorders and kidney disease. When all other socio-economic and geographical variables are taken into consideration, Veterans in 2019 are estimated to have higher rates of morbidity for obesity/overweight status, heart disease (despite the time series graphs and after accounting for other variables), stroke, skin cancer, cancer, COPD (with an impressive odds ratio of 1.461), arthritis, and diabetes. These results may indicate, again, that the multiple deployments our military personnel have experienced over the last decade have had a significant impact on their overall health once separated from the military [1,38].

### Strengths and Limitations

This study is the first to examine the various self-reported comorbidities among Veterans using the BRFSS dataset simultaneously with a robust statistical methodology. However, there are limitations. First, the BRFSS is cross-sectional, thus, odds ratios only provide the prevalence of each morbidity for that specific time period. Moreover, as illustrated in Table 2, the reporting of the morbidities was introduced at various years. Given the reliance of BRFSS of self-reporting, there is no feasible way to confirm the Veteran status for the respondents nor their medical diagnosis [45]. Furthermore, due to data limitation, this study was not able to include variables associated with military services, such as military rank, the number of deployments, the length of time in service, branch of service, the type of combat or exposure, the length of time between deployment and survey year, the number of injuries, the types of injuries, and the combat geographic locations. Future studies that include these variables should be considered. Finally, although there is complex weighting to account for sampling bias, there may still be issues related to coverage, selection bias, and an unrepresentative sample given that individuals within the Veteran population might be homeless and not included in the BRFSS (i.e., no cell phone or landline to be able to participate in the survey) [46].

## 5. Conclusions

The findings highlighted in this study paint a very concerning picture of the collective poor health of the US. Veteran. Our study findings suggest that when taken as a whole, these morbidities among Veterans could prompt the U.S. Department of Veteran Affairs (VA) to re-evaluate certain disabilities that the VA presumes were caused by military service (presumptive Service connection), as well as the impact of Veterans having multiple co-morbidities. Future studies should examine the interaction of multiple comorbidities on health outcomes of this population. Additionally, future studies should focus on possible reasons for the low occurrence of self-reported mental health disorders among those who served in the military. The findings in this study suggest a possible reluctance among Veterans to report these disorders and future studies should inquire as to potential causes. The study findings may also indicate an insufficient level of assessment among the Veteran population that accurately captures the true level of mental health challenges such as depression. Agencies such as the VA may potentially utilize these findings to develop and implement improved mental health assessment tools and procedures to better serve the Veteran population. Future studies should also seek to clarify the disproportionate impact and worsening health morbidities across targeted time periods for the Veteran group, either by stratifying our analysis or by generating a separate measure for multiple deployments and/or additional stressors as well as additional occupational requirement. Agencies such as the VA can use the findings of this study to better target scarce resources to geographies where there may be clusters of Veterans residing. The Veterans Health Administration (VHA), which is responsible for running the health system providing for the health needs of Veterans, may already be feeling the economic and resource impacts as a result of having an unhealthy Veteran population. However, additional research is needed using data sets that may contain more information that is specifically focused on the Veteran population. Such data may reside in certain organizations that cater to the needs of Veterans such as the American Legion, Veterans of Foreign Wars (VFW), or the Military Officers Association of America (MOAA).

Future studies might also examine the geographic locations where Veteran populations in the U.S. congregate [24]. These findings could assist the VA to design improved health intervention programs which could provide the necessary support to target populations. Additionally, this information could inform governmental and not-for-profit Veteran-serving institutions on the magnitude of the overall health needs of the Veteran population and help develop more effective health interventions aimed at improving the overall health of the Veteran. By resourcing these organizations, community interventions can be built to improve the health of the U.S. Veteran population: a population who sacrificed much in service to their country.

## Figures and Tables

**Figure 1 healthcare-09-00604-f001:**
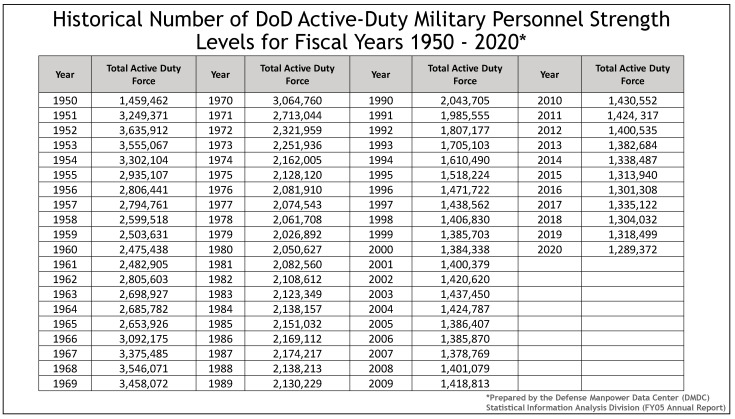
Total Personnel levels for U.S. Department of Defense (DoD) military personnel from 1950 through present (US Defense Manpower Data Center (DMDC)).

**Figure 2 healthcare-09-00604-f002:**
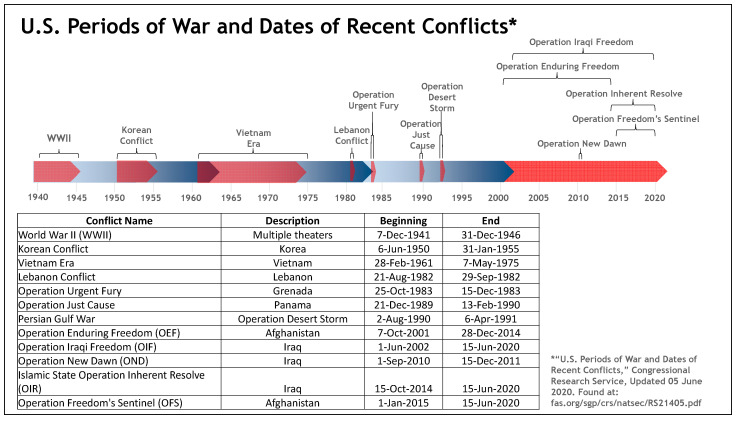
U.S. Periods of war and dates of recent conflicts from 1940 through present.

**Figure 3 healthcare-09-00604-f003:**
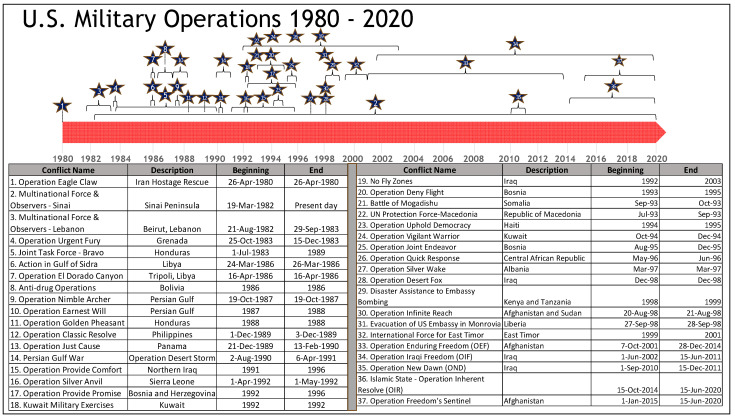
U.S. military operations from 1980 through present.

**Table 1 healthcare-09-00604-t001:** Population estimates and sample sizes by year and Veteran status.

Year	Non-Vet Population	Vet Population	Population Estimate	Non-Vet Sample	Vet Sample	Total Sample
2003	190,348,049	30,003,072	220,351,121	228,159	36,525	264,684
2004	191,637,278	29,746,086	441,734,485	260,982	42,840	303,822
2005	194,578,583	29,532,523	445,494,470	305,107	51,005	356,112
2006	198,138,945	29,118,914	451,368,965	304,989	50,721	355,710
2007	202,498,717	27,673,461	457,430,037	370,990	59,922	430,912
2008	205,615,985	27,244,684	463,032,847	358,433	56,076	414,509
2009	208,756,506	26,249,349	467,866,524	374,909	57,698	432,607
2010	211,037,577	26,048,662	472,092,094	390,643	60,432	451,075
2011	212,198,501	25,812,791	475,097,531	441,873	64,594	506,467
2012	216,959,427	26,098,283	481,069,002	415,817	59,870	475,687
2013	219,968,409	26,056,006	489,082,125	430,268	61,505	491,773
2014	220,704,167	27,778,365	494,506,947	402,544	62,120	464,664
2015	224,174,518	27,172,620	499,829,670	383,614	57,842	441,456
2016	227,144,466	27,006,670	505,498,274	422,384	63,919	486,303
2017	229,254,924	26,398,281	509,804,341	392,148	57,868	450,016
2018	230,694,063	27,379,324	513,726,592	381,382	56,054	437,436
2019	226,740,688	25,689,603	510,503,678	365,038	53,230	418,268
Totals	3,610,450,803	465,008,694	7,898,488,703	6,229,280	952,221	7,181,501

**Table 2 healthcare-09-00604-t002:** Variables in the Study.

Type	Name	
Dependent Variable: Overweight/Obese	_RFBMI5	Adults who have a body mass index greater than 25.00. No, Yes, Don’t Know/Refused/Missing
Dependent Variable: Angina or coronary heart disease	CVDCRHD#	(Ever told) you had angina or coronary heart disease? Yes, No, Don’t Know/Not Sure, Not Asked/Missing
Dependent Variable: Stroke	CVDSTRK#	(Ever told) you had a stroke. Yes, No, Don’t Know/Not Sure, Not Asked/Missing
Dependent Variable: Skin cancer	CHCSNCR	(Ever told) you had skin cancer? Yes, No, Don’t Know/Not Sure, Not Asked/Missing
Dependent Variable: Other cancer	CHCOCNCR	(Ever told) you had any other types of cancer? Yes, No, Don’t Know/Not Sure, Not Asked/Missing
Dependent Variable: COPD	CHCCOPD#	(Ever told) (you had) C.O.P.D., emphysema or chronic bronchitis? Yes, No, Don’t Know/Not Sure, Not Asked/Missing
Dependent Variable: Arthritis	HAVARTH#	Ever told) (you had) some form of arthritis, rheumatoid arthritis, gout, lupus, or fibromyalgia? Yes, No, Don’t Know/Not Sure, Not Asked/Missing
Dependent Variable: Mental Health	MENTHLTH	Now thinking about your mental health, which includes stress, depression, and problems with emotions, for how many days during the past 30 days was your mental health not good? Number of Days, None, Don’t Know/Not Sure, Refused, Not Asked/Missing
Dependent Variable: Kidney Disease	CHCKDNY	Not including kidney stones, bladder infection or incontinence, were you ever told you had kidney disease? Yes, No, Don’t Know/Not Sure, Not Asked/Missing
Dependent Variable: Diabetes	DIABETE#	(Ever told) (you had) diabetes? Yes, Yes Gestational Only, No, No Pre-Diabetes, Don’t Know/Not Sure, Refused, Not Asked/Missing (Recoded to Yes Non-Gestational, No, Unknown)
Demographic Control: Age	_AGE_G	Six-level imputed age category: 18–24, 25–34, 35–44, 45–54, 55–64, 65+
Demographic Control: Race	_IMPRACE	Imputed race/ethnicity value: White Non-Hispanic, Black Non-Hispanic, Asian Non-Hispanic, American Indian/Alaska Native Non-Hispanic, Hispanic, Other Race Non-Hispanic
Demographic Control: Gender	SEX	Calculated sex variable: Birth Sex Male, Birth Sex Female
Demographic Control: Marital Status	MARITAL	Are you: Married, Divorced, Widowed, Separated, Never Married, Unmarried Couple, Refused, Not Asked/Missing
Socioeconomic Control: Income	INCOME#	Is your annual household income from all sources: <$10 K, <$15 K, <$20 K, <$25 K, <$35 K, <$50 K, <$75 K, $75 K+, Don’t Know/Not Sure, Refused, Not Asked/Missing
Socioeconomic Control: Education	EDUCA	Level of education completed: < High School, Graduated High School, Attended College/Technical School, Graduated College/Technical School, Don’t Know/Not Sure/Missing
Socioeconomic Control: Employment	EMPLOY#	Are you currently…? Employed for Wages, Self-Employed, Out of Work 1+ Years, Out of Work <1 Year, A Homemaker, A Student, Retired, Unable to Work, Refused, Not Asked/Missing
Geographical control: State	_STATE	Federal Information Processing Standard Code for State
Independent Variable: Veteran Status	VETERAN#	Have you ever served on active duty in the United States Armed Forces, either in the regular military or in a National Guard or military reserve unit?
Weighting Variable	_STSTR	Sample Design Stratification Variable
Weighting Variable	_LLCPWT	Final weight assigned to each respondent

**Table 3 healthcare-09-00604-t003:** Year 2019 dependent variable proportions are shown (sample/population estimate).

Question *	Yes	No	Unknown
Overweight/Obese	62%/60%	29%/30%	9%/10% ^+^
Coronary Heart Disease	6%/4%	93%/95%	1%/1%
Stroke	4%/3%	95%/96%	1%/1%
Skin Cancer	10%/6%	90%/93%	0%/0%
Other Cancer	10%/7%	90%/93%	0%/0%
COPD	8%/7%	91%/93%	1%/0%
Arthritis	33%/25%	66%/75%	1%/0%
Mental Health	34%/37%	66%/63%	0%/0%
Kidney Issues	4%/3%	96%/97%	0%/0%
Diabetes	14%/11%	86%/89%	0%/0%

* May not add to 100% due to rounding, ^+^ maximum missing of any year/variable.

**Table 4 healthcare-09-00604-t004:** Starting years for analysis.

Variable	Year
Overweight/Obese	2003
Heart Disease	2005
Stroke	2005
Skin Cancer	2011
Cancer	2011
COPD	2011
Arthritis	2011
Mental Health	2003
Kidney Disease	2011
Diabetes	2003

**Table 5 healthcare-09-00604-t005:** Age-adjusted comparisons for all dependent variables for all years by Veteran and non-Veteran status.

	Obesity	Diabetes	Mental Health	Heart Disease	Stroke	Skin Cancer	Cancer	COPD	Arthritis	Kidney Disease
Year *	Non-Vet	Vet	Non-Vet	Vet	Non-Vet	Vet	Non-Vet	Vet	Non-Vet	Vet	Non-Vet	Vet	Non-Vet	Vet	Non-Vet	Vet	Non-Vet	Vet	Non-Vet	Vet
Y2003	0.56	0.70	0.11	0.13	0.37	0.31	N/A	N/A	N/A	N/A	N/A	N/A	N/A	N/A	N/A	N/A	N/A	N/A	N/A	N/A
Y2004	0.57	0.70	0.12	0.13	0.37	0.31	N/A	N/A	N/A	N/A	N/A	N/A	N/A	N/A	N/A	N/A	N/A	N/A	N/A	N/A
Y2005	0.58	0.70	0.13	0.14	0.36	0.30	0.08	0.13	0.05	0.06	N/A	N/A	N/A	N/A	N/A	N/A	N/A	N/A	N/A	N/A
Y2006	0.58	0.71	0.13	0.15	0.37	0.30	0.08	0.14	0.05	0.06	N/A	N/A	N/A	N/A	N/A	N/A	N/A	N/A	N/A	N/A
Y2007	0.59	0.72	0.14	0.16	0.36	0.28	0.08	0.13	0.05	0.06	N/A	N/A	N/A	N/A	N/A	N/A	N/A	N/A	N/A	N/A
Y2008	0.60	0.73	0.14	0.15	0.36	0.30	0.08	0.14	0.05	0.06	N/A	N/A	N/A	N/A	N/A	N/A	N/A	N/A	N/A	N/A
Y2009	0.60	0.73	0.14	0.16	0.36	0.30	0.07	0.13	0.05	0.06	N/A	N/A	N/A	N/A	N/A	N/A	N/A	N/A	N/A	N/A
Y2010	0.61	0.73	0.14	0.16	0.36	0.30	0.08	0.14	0.05	0.06	N/A	N/A	N/A	N/A	N/A	N/A	N/A	N/A	N/A	N/A
Y2011	0.60	0.73	0.15	0.18	0.37	0.30	0.08	0.13	0.05	0.06	0.11	0.16	0.11	0.13	0.08	0.09	0.37	0.36	0.03	0.03
Y2012	0.60	0.72	0.15	0.18	0.37	0.30	0.08	0.13	0.05	0.06	0.11	0.16	0.11	0.12	0.08	0.09	0.38	0.38	0.04	0.04
Y2013	0.60	0.72	0.16	0.18	0.35	0.29	0.08	0.13	0.05	0.06	0.11	0.16	0.11	0.16	0.08	0.10	0.38	0.37	0.04	0.04
Y2014	0.59	0.72	0.16	0.18	0.35	0.29	0.08	0.13	0.05	0.06	0.11	0.17	0.11	0.12	0.09	0.10	0.39	0.37	0.04	0.04
Y2015	0.59	0.71	0.16	0.19	0.36	0.30	0.07	0.12	0.05	0.06	0.12	0.17	0.12	0.13	0.08	0.10	0.37	0.37	0.04	0.04
Y2016	0.59	0.71	0.17	0.19	0.36	0.30	0.08	0.12	0.05	0.06	0.11	0.16	0.11	0.13	0.08	0.10	0.38	0.38	0.04	0.05
Y2017	0.59	0.71	0.17	0.19	0.38	0.31	0.07	0.12	0.05	0.06	0.12	0.17	0.12	0.14	0.09	0.11	0.37	0.37	0.04	0.05
Y2018	0.60	0.72	0.17	0.20	0.39	0.32	0.07	0.12	0.05	0.07	0.12	0.17	0.12	0.14	0.09	0.11	0.38	0.39	0.05	0.05
Y2019	0.60	0.71	0.17	0.19	0.41	0.35	0.07	0.11	0.05	0.07	0.12	0.18	0.12	0.14	0.09	0.11	0.37	0.37	0.05	0.05
Y11-Y19	0.60	0.72	0.16	0.19	0.37	0.30	0.08	0.13	0.05	0.06	0.11	0.16	0.11	0.13	0.09	0.10	0.38	0.37	0.04	0.04
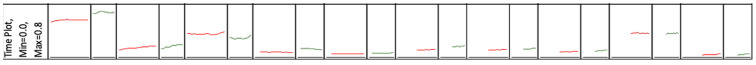

* Data from 2003 to 2010 were not combined due to differences in weighting and data-collection by the US Centers for Disease Control and Prevention (CDC). Data for some of the variables were not collected by the CDC until 2011.

**Table 6 healthcare-09-00604-t006:** Age-adjusted comparisons for all dependent variables for years 2011–2019 by Veteran and non-Veteran status.

Variable	Overweight/Obese	Heart Disease	Stroke	Skin Cancer	Cancer	COPD	Arthritis	Mental Health	Kidney Disease	Diabetes
(Intercept)	0.478 ***	0.004 ***	0.007 ***	0.001 ***	0.009 ***	0.035 ***	0.055 ***	1.356 ***	0.012 ***	0.015 ***
25 to 34	2.010 ***	2.074 ***	2.753 ***	1.504 ***	2.663 ***	1.834 ***	2.87 ***	0.915 ***	1.728 ***	2.402 ***
35 to 44	2.827 ***	4.645 ***	5.724 ***	3.518 ***	4.340 ***	2.863 ***	6.354 ***	0.844 ***	2.765 ***	6.931 ***
45 to 54	3.305 ***	11.432 ***	10.822 ***	8.587 ***	7.742 ***	4.831 ***	13.269 ***	0.748 ***	4.158 ***	15.439 ***
55 to 64	3.526 ***	21.610 ***	15.455 ***	15.812 ***	12.604 ***	6.285 ***	22.293 ***	0.587 ***	5.744 ***	25.632 ***
65 or older	3.111 ***	32.13 ***	19.386 ***	33.477 ***	21.72 ***	5.484 ***	27.973 ***	0.283 ***	7.123 ***	30.954 ***
Caucasian	0.960 ***	1.128 ***	0.768 ***	6.149 ***	1.342 ***	1.291 ***	1.205 ***	1.235 ***	0.864 ***	0.596 ***
Hispanic	1.155 ***	0.906 ***	0.655 ***	1.171 ***	0.812 ***	0.653 ***	0.693 ***	0.903 ***	1.033	1.013
Male	1.889 ***	1.676 ***	1.093 ***	1.058 ***	0.618 ***	0.816 ***	0.664 ***	0.648 ***	0.948 ***	1.212 ***
Married	1.072 ***	0.877 ***	0.718 ***	1.086 ***	0.953 ***	0.643 ***	0.850 ***	0.677 ***	0.833 ***	0.943 ***
Income ≥ $75K	1.044 ***	0.749 ***	0.579 ***	1.180 ***	0.953 ***	0.515 ***	0.794 ***	0.836 ***	0.759 ***	0.694 ***
College Graduate	0.689 ***	0.739 ***	0.639 ***	1.249 ***	1.016 ^+^	0.473 ***	0.671 ***	0.949 ***	0.789 ***	0.661 ***
Employed for Wages	1.186 ***	0.472 ***	0.351 ***	0.78 ***	0.651 ***	0.46 ***	0.613 ***	0.756 ***	0.473 ***	0.663 ***
East North Central	1.076 ***	1.217 ***	1.283 ***	1.48 ***	1.037 *	1.297 ***	1.157 ***	0.882 ***	1.016	1.203 ***
East South Central	0.817 ***	0.999	0.869 ***	0.977	0.986	0.887 ***	0.899 ***	0.928 ***	0.839 ***	0.934 ***
Middle Atlantic	0.787 ***	0.802 ***	0.885 ***	1.476 ***	0.994	0.840 ***	0.849 ***	0.961 ***	1.039 ^+^	0.832 ***
Mountain	0.790 ***	0.890 ***	0.840 ***	1.062 ***	1.039 **	0.870 ***	0.894 ***	0.976 **	0.877 ***	0.892 ***
Pacific	0.743 ***	0.843 ***	0.845 ***	1.417 ***	1.027	0.766 ***	0.790 ***	1.035 ***	0.968	0.883 ***
South Atlantic	0.883 ***	1.022	1.027	1.562 ***	0.996	1.022 ^+^	0.905 ***	0.853 ***	0.982	0.990
West North Central	0.986	1.975 ***	0.597 ***	0.89 *	0.902 **	0.785 ***	1.025	0.553 ***	0.745 ***	0.976
West South Central	0.961 ***	0.913 ***	0.976	1.067 ***	0.998	0.849 ***	0.849 ***	0.808 ***	0.86 ***	0.935 ***
Territories of U.S	0.980 ^+^	1.052*	1.107 ***	1.306 ***	0.979	0.962 *	0.873 ***	0.845 ***	1.044	1.08 ***
Veteran	1.239 ***	1.34 ***	1.276 ***	1.302 ***	1.509 ***	1.355 ***	1.247 ***	0.972 **	1.152 ***	1.108 ***
Year 2012	1.009	1.048 **	0.988	0.961 **	0.976	1.029 ^+^	1.069 ***	1.011	1.061 *	1.04 **
Year 2013	1.006	0.956 *	0.998	0.999	1.006	1.049 **	1.022 *	0.939 ***	1.047 ^+^	1.04 **
Year 2014	0.974 **	0.992	1.045 *	0.979	0.974 ^+^	1.069 ***	1.054 ***	0.941 ***	1.093 ***	1.066 ***
Year 2015	0.956 ***	0.904 ***	1.029	1.000	1.003	1.019	0.988	0.97 **	1.054 *	1.055 ***
Year 2016	0.960 ***	0.965 *	1.076 **	0.990	0.988	1.067 ***	1.034 ***	0.982 *	1.163 ***	1.088 ***
Year 2017	0.982 *	0.89 ***	1.087 ***	0.989	1.018	1.076 ***	0.972 **	1.043 ***	1.243 ***	1.102 ***
Year 2018	1.000	0.958 *	1.156 ***	1.013	1.020	1.142 ***	1.056 ***	1.076 ***	1.224 ***	1.165 ***

*** *p* < 0.001, ** *p* < 0.01, * *p* < 0.05, ^+^
*p* < 0.10.

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
