# Peer review of "Exploring Health Outcomes for U.S. Veterans Compared to Non-Veterans from 2003 to 2019"

_healthcare, 2021, doi:10.3390/healthcare9050604_

Round 1
Reviewer 1 Report
Dear Editor,
Thank you for inviting me to review this manuscript. The aim of this study was to better understand how the U.S Veterans health outcomes compared to non-Veterans. This study utilized the Behavioral Risk Factor Surveillance System (BRFSS) dataset from the period 2003 through 2019, to examine 10 different self-reported morbidities including overweight / obesity, heart disease, stroke, skin cancer, cancer, COPD, arthritis, mental health, kidney disease, and diabetes. I believe that the topic is very interesting with scientific interest and that the authors put effort in drafting this work. However, the current version of the manuscript is let down by its written presentation, the results are not well presented and important details in the methods are lacking or not clear. Nevertheless, I offer comments and suggestions for the author’s consideration.
Introduction
The authors will need to spend time with the Introduction of their manuscript to describe the concept of “multiple chronic conditions” and the possible associations with behavioral risk factors. To help I would strongly recommend to look at the following references that also utilized the BRSFF data.
- Adams, M. (2019). The impact of key modifiable risk factors on leading chronic conditions. Preventive Medicine., 120, 113–118. https://doi.org/10.1016/j.ypmed.2019.01.006
- Adams, M. (2017). Linear association between number of modifiable risk factors and multiple chronic conditions: Results from the Behavioral Risk Factor Surveillance System. Preventive Medicine., 105, 169–175. https://doi.org/10.1016/j.ypmed.2017.09.013
- Stone C. L. (2020). A population-based measure of chronic disease severity for health planning and evaluation in the United States. AIMS public health, 7(1), 44–65. https://doi.org/10.3934/publichealth.2020006
Methods
General comment: The methods section is a bit confusing and hard to follow. To help the reader it would be recommended to re-structure the section. Recommendation to re-structure: Data source ->Study Sample-> Institutional Review Board (IRB) -> Study Measures: Dependent Variable, Independent Variables, Study covariates-> Data Analysis. In addition, some additional details and clarification of the methods used would be helpful.
- Data: To demonstrate the strength of the BRSFF data. It would be recommended to expend the data sub-section. For example, how is the data collected? How many interviews per year are collected? In addition, there should be explanation regarding the BRSFF questionnaire protocol.
- The authors have stated: “The BRFSS is a continuous, surveillance system that collects information from each state in the U.S. about modifiable risk factors for chronic diseases and other leading causes of death [22]”. However, the BRFSS now collects data in all 50 states as well as the District of Columbia and 3 U.S. territories.
- The authors have also stated: “Each year was used to estimate the population point estimate for the specific time and morbidity. These data are considered by the CDC to be the nation’s best source for health-related survey data, and the 2019 version includes 418,268 observations. Applying complex weighting to these observations results in a picture of the entire nation’s health self-reported health [23].” First, I would recommend that this section should be moved under sampling. Second, it is not clear what the authors mean “complex weighting”. In 2011, BRFSS began using a weighting method known as iterative proportional fitting, or raking. Could the authors please clarify the sampling section?
- Institutional Review Board- Currently the manuscript is missing IRB statement the authors should address this.
- Software: I would recommend that this section should be moved under Data Analysis.
- Sample Size: It isn’t clear how the sample was construct and the authors should expend the section
- The authors have stated: “Table 1 shows the sample size for each of the years in question” - Could the authors clarify what table 1 present? Is it weighted percentage or unweighted sample size? Is the sample include both veteran and non-veteran?
- The authors also stated “ Complex weights were applied to estimate the entire population, resulting in fractional values”. – Yet it is not clear how “complex weight were applied”. There should be clear explanation regarding how the 2003-2019 survey data were combined, weighted percentages was calculated what for what variables and which groups
- The independent variable: It is stated: “The primary variables of interest were the year of the estimate (2003 through 2018), and Veteran status, coded {0 = non-Veteran, 1= Veteran} with blank, refused, and do not know coded as non-Veteran”
- Currently it isn’t clear how veteran status variable was obtained for this study?
- Based on what question/ information from the BRSFF data base was it constructed?
- Regarding the re-coding “blank, refused, and do not know coded as non-Veteran”
- Blank are typically state that were not asked the questions, refused and don’t know are consider missing data. First, it is recommended to report the numbers; in addition typically those responses are excluded from the analysis. It is not clear way it was re-coded as non-Veteran. Could the authors justify the decision?
- Dependent variables: It is stated: “Dependent variables in the study were dichotomously coded (0 = absence of morbidity, 1 = presence of morbidity) and included overweight / obese, heart disease, stroke, skin cancer, cancer, COPD, arthritis, mental health condition, kidney disease, and diabetes. In all cases, “do not know”, “refused”, “unknown”, and “missing” responses were coded 0 (not present).”
- Currently it isn’t clear how morbidity status was obtained for this study?
- Based on what questions/ information from the BRSFF database was it constructed?
- Regarding the re-coding ““do not know”, “refused”, “unknown”, and “missing” responses were coded 0 (not present)” – First it is recommended to report the numbers; in addition typically those responses are excluded from the analysis. It is not clear way it was re-coded as non- present. Could the authors justify the decision?
- The authors’ state: “Many of the dependent variables did not have observations dating back to 2003. Table 2 shows the analysis starting year for each of the variables.” - However could the authors please clarify did the information was collected across all 50 states as well as the District of Columbia and 3 U.S. territories
- The authors combined data analysis and other potential variables under dependent Which is extremely confusing and hard to follow. It would be recommended that the authors would create two separate subheading. For example first subheading: Study Covariates for example: here included race/ethnicity (white, black and other race/ethnicity), age (18–24, 25–34, 35–44, 45–54, 55–64) etc... In addition, it would recommend excluding the SASS variable code for example “_IMPRACE”. The second subheading: Data Analysis- that include clear description of data analysis that been carried in the study. That isn’t clear.
Results
It isn’t clear -How much participants included in the study; how much was in the veterans ? how much was in the non-veterans , how much was in each chronic conditions. The authors should consider present table of weighted percentages for demographics and chronic conditions variables by veteran and non-veteran
Discussion
The author’s state: “Multiple deployments over the last several decades may be a major contributor to increased illness seen after separation from the service [30]. The demand placed upon the U.S. Service member, particularly since the mid to late 1980’s has been illustrated in the number of military operations requiring deployment [20]. Unfortunately, the number of available personnel to support these deployments has not kept pace, resulting in more Service members deploying more often. As depicted in the health outcomes data, this combination of increased demand in military operations placed upon a smaller population may be resulting in a group of at-risk individuals for a variety of debilitating health conditions [1]. Additional stressors may exist in the occupational requirement for all Service members to meet stringent height and weight standards, and physical fitness standards, both of which are evaluated every six months [17]. Perhaps the combination of each of these stressors ultimately contribute to the stark differences in health outcomes between the Veteran populations compared to non-Veteran populations [1-3] As depicted in the health outcomes data, this combination of increased demand in military operations placed upon a smaller population may be resulting in a group of at-risk individuals for a variety of debilitating health conditions [1]. Additional stressors may exist in the occupational requirement for all Service members to meet stringent height and weight standards, and physical fitness standards, both of which are evaluated every six months [17].”
Yet the study didn’t examine any of the variables [Information on deployment or height and weight standards, and physical fitness standards] the authors cannot provide data. This section should be remove.
Author Response
Thank you for giving us the opportunity to submit a revised draft of the manuscript, “Exploring Health Outcomes for U.S. Veterans Compared to Non-Veterans from 2003 to 2019“ for publication in Healthcare. We appreciate the time and effort that you and your reviewers dedicated to providing feedback on our manuscript and are grateful for the insightful comments on the valuable improvements to our paper. We have incorporated the recommendations made by the reviewers. These changes are highlighted within the manuscript for your ease in reading. Please see below (in blue text), for a point-by-point response to the reviewers’ comments and concerns. All page numbers refer to the revised manuscript file with tracked changes.
Reviewer comment 1: Thank you for inviting me to review this manuscript. The aim of this study was to better understand how the U.S Veterans health outcomes compared to non-Veterans. This study utilized the Behavioral Risk Factor Surveillance System (BRFSS) dataset from the period 2003 through 2019, to examine 10 different self-reported morbidities including overweight / obesity, heart disease, stroke, skin cancer, cancer, COPD, arthritis, mental health, kidney disease, and diabetes. I believe that the topic is very interesting with scientific interest and that the authors put effort in drafting this work.
Author response: Thank you for your kinds comments. Yes, our Study Team saw a need to address this very important topic for a population that has sacrificed so much.
However, the current version of the manuscript is let down by its written presentation, the results are not well presented and important details in the methods are lacking or not clear. Nevertheless, I offer comments and suggestions for the author’s consideration.
Author response: We truly appreciate your valuable assistance in improving this important manuscript. We have embraced your concerns and recommended modifications and have added a new online reference that provides all of our variable encoding / recoding / feature engineering as well as all of our analysis for replication.
“To compare Veteran and non-Veteran health outcomes, yearly age-adjusted, population estimates from 2003 (or earliest year available) to 2019, were used for Veteran / non-Veteran. The entirety of the analysis is available for review at the following location https://rpubs.com/R-Minator/BRFSS.”
Reviewer comment 2: Introduction
The authors will need to spend time with the Introduction of their manuscript to describe the concept of “multiple chronic conditions” and the possible associations with behavioral risk factors. To help I would strongly recommend to look at the following references that also utilized the BRSFF data.
- Adams, M. (2019). The impact of key modifiable risk factors on leading chronic conditions. Preventive Medicine., 120, 113–118. https://doi.org/10.1016/j.ypmed.2019.01.006
- Adams, M. (2017). Linear association between number of modifiable risk factors and multiple chronic conditions: Results from the Behavioral Risk Factor Surveillance System. Preventive Medicine., 105, 169–175. https://doi.org/10.1016/j.ypmed.2017.09.013
- Stone C. L. (2020). A population-based measure of chronic disease severity for health planning and evaluation in the United States. AIMS public health, 7(1), 44–65. https://doi.org/10.3934/publichealth.2020006
Author response: Thank you for your insightful recommendations. We have added these recommended references.
Reviewer comment 3: Methods
General comment: The methods section is a bit confusing and hard to follow. To help the reader it would be recommended to re-structure the section. Recommendation to re-structure: Data source ->Study Sample-> Institutional Review Board (IRB) -> Study Measures: Dependent Variable, Independent Variables, Study covariates-> Data Analysis. In addition, some additional details and clarification of the methods used would be helpful.
Author response: Thank you for your comment and recommendations. Based on all of your comments, we have restructured as follows.
- Materials and Methods
2.1. Data Source
2.2. Study Sample
2.3. Study Measures
2.3.1. Dependent Variables
2.3.2. Independent Variable
2.3.3. Covariates
2.3.3.1. Demographic Variables
2.3.3.2. Socio-Economic Variables
2.3.3.3. Geographic Variables
2.3.3.4. Time Variable
2.4. Methods & Models
2.4.1. Descriptive Models
2.4.2. General Linear Models
2.4.3. Data Analysis
- Results
3.1. Descriptive Statistics
3.2. General Linear Models
- Data: To demonstrate the strength of the BRSFF data. It would be recommended to expand the data sub-section. For example, how is the data collected? How many interviews per year are collected? In addition, there should be explanation regarding the BRSFF questionnaire protocol.
Author response: Thank you very much for your comments regarding the structure and information needed for the methods section. Our Study Team re-structured this section based on your recommendation to help with the flow of the information. Regarding the data source, we included additional information regarding the methods of data collections for the BRFSS, where it is collected and also added in an additional reference to the CDC website where a robust methodology on the BRFSS can be readily found. We also provided an updated Table 1 that illustrates the sheer size of the samples.
2.2. Study Sample
For this study, BRFSS data from 2003 to 2019 (last available year) were acquired from the CDC [21]. Each year was used to estimate the population point estimate for the specific time and morbidity. The BRFSS data are considered by the CDC to be the nation’s best source for health-related survey data, and the 2019 version includes 418,268 observations. Applying stratum and individual weights to these observations results in a picture of the entire nation’s health self-reported health [23]. En toto, this study included 7.181 million sample observations.
Table 1 shows the population estimates and sample sizes for each of the years in question by Veteran / Non-Veteran status. Sampling weights based on location and socio-demographic information were applied to estimate the entire population, resulting in fractional values for population estimates. Prior to 2011, samples were weighted by stratum (area code / prefix combinations) times the inverse number of telephones in the household times the number of adults in the household times the number of people in an age-by-sex or age-by-race/ethnicity-by-sex category in the population of a region or a state divided by the sum of the preceding weights for the respondents in the same age-by-sex or age-by-race/ethnicity-by-sex category (adjusting for noncoverage and nonresponse and forces the sum of the weighted frequencies to equal population estimates for the region or state.) Two separate variable weight fields. This weighting was applied through the use of separate fields for stratum and weights. From 2011 onwards, raking replaced the post-stratification process. Raking allows for incorporation of cellular telephone survey data and allows for inclusion of other demographic considerations to improve socio-demographic matching by stratum.
Table 1. Population estimates and sample sizes by year and veteran status
Year |
Non-Vet Population |
Vet Population |
Population Estimate |
Non-Vet Sample |
Vet Sample |
Sample |
2003 |
190,348,049 |
30,003,072 |
220,351,121 |
228,159 |
36,525 |
264,684 |
2004 |
191,637,278 |
29,746,086 |
441,734,485 |
260,982 |
42,840 |
303,822 |
2005 |
194,578,583 |
29,532,523 |
445,494,470 |
305,107 |
51,005 |
356,112 |
2006 |
198,138,945 |
29,118,914 |
451,368,965 |
304,989 |
50,721 |
355,710 |
2007 |
202,498,717 |
27,673,461 |
457,430,037 |
370,990 |
59,922 |
430,912 |
2008 |
205,615,985 |
27,244,684 |
463,032,847 |
358,433 |
56,076 |
414,509 |
2009 |
208,756,506 |
26,249,349 |
467,866,524 |
374,909 |
57,698 |
432,607 |
2010 |
211,037,577 |
26,048,662 |
472,092,094 |
390,643 |
60,432 |
451,075 |
2011 |
212,198,501 |
25,812,791 |
475,097,531 |
441,873 |
64,594 |
506,467 |
2012 |
216,959,427 |
26,098,283 |
481,069,002 |
415,817 |
59,870 |
475,687 |
2013 |
219,968,409 |
26,056,006 |
489,082,125 |
430,268 |
61,505 |
491,773 |
2014 |
220,704,167 |
27,778,365 |
494,506,947 |
402,544 |
62,120 |
464,664 |
2015 |
224,174,518 |
27,172,620 |
499,829,670 |
383,614 |
57,842 |
441,456 |
2016 |
227,144,466 |
27,006,670 |
505,498,274 |
422,384 |
63,919 |
486,303 |
2017 |
229,254,924 |
26,398,281 |
509,804,341 |
392,148 |
57,868 |
450,016 |
2018 |
230,694,063 |
27,379,324 |
513,726,592 |
381,382 |
56,054 |
437,436 |
2019 |
226,740,688 |
25,689,603 |
510,503,678 |
365,038 |
53,230 |
418,268 |
Totals |
3,610,450,803 |
465,008,694 |
7,898,488,703 |
6,229,280 |
952,221 |
7,181,501 |
- The authors have stated: “The BRFSS is a continuous, surveillance system that collects information from each state in the U.S. about modifiable risk factors for chronic diseases and other leading causes of death [22]”.However, the BRFSS now collects data in all 50 states as well as the District of Columbia and 3 U.S. territories.
Author response: Thank you as you are absolutely correct. Based on your recommendation, we corrected this discussion and actually use a factor-level indicating “Territory” in our analysis.
The BRFSS is a continuous health survey system that collects information from each state in the U.S., three U.S. territories and the District of Columbia about modifiable risk factors for chronic diseases and other leading causes of death [22].
- The authors have also stated: “Each year was used to estimate the population point estimate for the specific time and morbidity. These data are considered by the CDC to be the nation’s best source for health-related survey data, and the 2019 version includes 418,268 observations. Applying complex weighting to these observations results in a picture of the entire nation’s health self-reported health [23].”First, I would recommend that this section should be moved under sampling. Second, it is not clear what the authors mean “complex weighting”. In 2011, BRFSS began using a weighting method known as iterative
Author response: Thank you for your recommendation. Based on your recommendation, we now have a separate section for study sample discussion (see above) and discuss the raking method from 2011 as well in 2.1.4.
From 2011 onwards, raking replaced the post-stratification process. Raking allows for incorporation of cellular telephone survey data and allows for inclusion of other demo-graphic considerations to improve socio-demographic matching by stratum.
Reviewer comment 4: Institutional Review Board- Currently the manuscript is missing IRB statement the authors should address this.
Author response: Thank you for calling this to our attention. BRFSS data is a publicly available, anonymous data set that does not require IRB review or an exempt status for its use. The data is posted to the CDC’s website. The following statement was added into the paper for clarity.
BRFSS data is a publicly available, anonymous data set that does not require institutional review board or an exempt status for its use.
Reviewer comment 5: Software: I would recommend that this section should be moved under Data Analysis.
Author response: Our Study Team fully appreciates this suggestion and we have moved this section under the title of Data Analysis. Thank you.
Reviewer comment 6: Sample Size: It isn’t clear how the sample was construct and the authors should expend the section
- The authors have stated: “Table 1 shows the sample size for each of the years in question” - Could the authors clarify what table 1 present? Is it weighted percentage or unweighted sample size? Is the sample include both veteran and non-veteran?
Author response: Table 1 was revamped to show the sample sizes and population estimates.
Table 1 shows the population estimates and sample sizes for each of the years in question by Veteran / Non-Veteran status. Sampling weights based on location and socio-demographic information were applied to estimate the entire population, resulting in fractional values for population estimates. Prior to 2011, samples were weighted by stratum (area code / prefix combinations) times the inverse number of telephones in the household times the number of adults in the household times the number of people in an age-by-sex or age-by-race/ethnicity-by-sex category in the population of a region or a state divided by the sum of the preceding weights for the respondents in the same age-by-sex or age-by-race/ethnicity-by-sex category (adjusting for noncoverage and nonresponse and forces the sum of the weighted frequencies to equal population estimates for the region or state.) Two separate variable weight fields. This weighting was applied through the use of separate fields for stratum and weights. From 2011 onwards, raking replaced the post-stratification process. Raking allows for incorporation of cellular telephone survey data and allows for inclusion of other demographic considerations to improve socio-demographic matching by stratum.
Table 1. Population estimates and sample sizes by year and veteran status
Year |
Non-Vet Population |
Vet Population |
Population Estimate |
Non-Vet Sample |
Vet Sample |
Sample |
2003 |
190,348,049 |
30,003,072 |
220,351,121 |
228,159 |
36,525 |
264,684 |
2004 |
191,637,278 |
29,746,086 |
441,734,485 |
260,982 |
42,840 |
303,822 |
2005 |
194,578,583 |
29,532,523 |
445,494,470 |
305,107 |
51,005 |
356,112 |
2006 |
198,138,945 |
29,118,914 |
451,368,965 |
304,989 |
50,721 |
355,710 |
2007 |
202,498,717 |
27,673,461 |
457,430,037 |
370,990 |
59,922 |
430,912 |
2008 |
205,615,985 |
27,244,684 |
463,032,847 |
358,433 |
56,076 |
414,509 |
2009 |
208,756,506 |
26,249,349 |
467,866,524 |
374,909 |
57,698 |
432,607 |
2010 |
211,037,577 |
26,048,662 |
472,092,094 |
390,643 |
60,432 |
451,075 |
2011 |
212,198,501 |
25,812,791 |
475,097,531 |
441,873 |
64,594 |
506,467 |
2012 |
216,959,427 |
26,098,283 |
481,069,002 |
415,817 |
59,870 |
475,687 |
2013 |
219,968,409 |
26,056,006 |
489,082,125 |
430,268 |
61,505 |
491,773 |
2014 |
220,704,167 |
27,778,365 |
494,506,947 |
402,544 |
62,120 |
464,664 |
2015 |
224,174,518 |
27,172,620 |
499,829,670 |
383,614 |
57,842 |
441,456 |
2016 |
227,144,466 |
27,006,670 |
505,498,274 |
422,384 |
63,919 |
486,303 |
2017 |
229,254,924 |
26,398,281 |
509,804,341 |
392,148 |
57,868 |
450,016 |
2018 |
230,694,063 |
27,379,324 |
513,726,592 |
381,382 |
56,054 |
437,436 |
2019 |
226,740,688 |
25,689,603 |
510,503,678 |
365,038 |
53,230 |
418,268 |
Totals |
3,610,450,803 |
465,008,694 |
7,898,488,703 |
6,229,280 |
952,221 |
7,181,501 |
- The authors also stated “ Complex weights were applied to estimate the entire population, resulting in fractional values”. – Yet it is not clear how “complex weight were applied”. There should be clear explanation regarding how the 2003-2019 survey data were combined, weighted percentages was calculated what for what variables and which groups
Author response: Thank you for your valuable recommendation. We previously referenced the reader to the BRFSS site for this information but have added the following statements to address your concerns. We also show how the weights are used to provide population estimates in Table 1. We are hopeful that these actions have addressed your concerns.
Complex weights based on location and socio-demographic information were applied to estimate the entire population, resulting in fractional values. Prior to 2011, samples were weighted by stratum (area code / prefix combinations) times the inverse number of telephones in the household times the number of adults in the household times the number of people in an age-by-sex or age-by-race/ethnicity-by-sex category in the population of a region or a state divided by the sum of the preceding weights for the respondents in the same age-by-sex or age-by-race/ethnicity-by-sex category (adjusting for noncoverage and nonresponse and forces the sum of the weighted frequencies to equal population estimates for the region or state.) Two separate variable weight fields. This weighting was applied through the use of separate fields for stratum and weights. From 2011 onwards, raking replaced the post-stratification process. Raking allows for incorporation of cellular telephone survey data and allows for inclusion of other demographic considerations to improve so-cio-demographic matching by stratum.
Reviewer comment 7: The independent variable: It is stated: “The primary variables of interest were the year of the estimate (2003 through 2018), and Veteran status, coded {0 = non-Veteran, 1= Veteran} with blank, refused, and do not know coded as non-Veteran”
- Currently it isn’t clear how veteran status variable was obtained for this study?
Author response: Thank you for your obvservation and question. You certainly bring up a good point and as a result, our study team now has both a table of variables, associated definitions, and BRFSS question.
The primary variables of interest were the year of the estimate (2003 through 2019), and Veteran status, recoded {0 = did not self-report as veteran, 1= self-reported as veteran}. This coding results from the survey questions that follow.
- Years 2003 through 2006 (Code: VETERAN). Have you ever served on active duty in the United States Armed Forces, either in the regular military or in a National Guard or military reserve unit?” 1=Yes, 2=No, 7=Don’t Know / Not Sure, 9=Refused, Blank=Not asked / Missing
- Years 2007 through 2008 (Code: VETERAN1). “Have you ever served on active duty in the United States Armed Forces, either in the regular military or in a National Guard or military reserve unit? Active duty does not include training for the Reserves or National Guard, but DOES include activation, for example, for the Persian Gulf War.” 1=Yes, 2=No, 7=Don’t Know / Not Sure, 9=Refused, Blank=Not asked / Missing
- Years 2009 (Code VETERAN2). “Have you ever served on active duty in the United States Armed Forces, either in the regular military or in a National Guard or military reserve unit? Active duty does not include training for the Reserves or National Guard, but DOES include activation, for example, for the Persian Gulf War.” 1=Yes, now on Active Duty, 2=Yes, on Active Duty during the last 12 months but not now, 3=Yes, on active duty in the past, but not during the last 12 months, 4=No, training for Reserves or National Guard only, 5=No, never served in the military, 7=Don’t Know / Not Sure, 9=Refused, Blank=Not asked / Missing
- Years 2010 through 2019 (Code: VETERAN3). “Have you ever served on active duty in the United States Armed Forces, either in the regular military or in a National Guard or military reserve unit? Active duty does not include training for the Reserves or National Guard, but DOES include activation, for example, for the Persian Gulf War.” 1=Yes, 2=No, 7=Don’t Know / Not Sure, 9=Refused, Blank=Not asked / Missing
In all years, the responses were identically coded with the exception of 2009. Recoding for this year assigned any yes values equal to 1 and all other values equal to zero. Due to the exceedingly low numbers of responses that were blank, refused, and “do not know” (e.g., 0.574% in 2019), these were imputed with the modal response of non-Veteran (86.7% of the population in 2019).
- Based on what question/ information from the BRSFF data base was it constructed?
Author response: Please see above. We also now include a new Table to define all variables for the base year definition for coding purposes. Thank you so much for your question.
- Regarding the re-coding “blank, refused, and do not know coded as non-Veteran” Blank are typically state that were not asked the questions, refused and don’t know are consider missing data. First, it is recommended to report the numbers; in addition typically those responses are excluded from the analysis. It is not clear way it was re-coded as non-Veteran. Could the authors justify the decision?
Author response: This is a great question and one that we certainly should have addressed in the manuscript. The proportion in these categories are negligible (a fraction of a percent) and might have been omitted; however, we preferred to impute them as the modal category (which is probabilistically non-veteran). Based on your question, our Study Team has added the following.
Due to the exceedingly low numbers of responses that were blank, refused, and “do not know” (e.g., 0.574% in 2019), these were imputed with the modal response of non-Veteran (86.7% of the population in 2019).
We also now demonstrate the unknown / refused missing fractions in the paper. Only in the case of overweight / obese reporting are the values greater than 1%. Thus we coded 1=positively identified as having X versus 0=did not positively identify as having X.
Table 3. Year 2019 dependent variable proportions are shown (sample / population estimate).
Question * |
Yes |
No |
Unknown |
Overweight / Obese |
62%/60% |
29%/30% |
9%/10% |
Coronary Heart Disease |
6%/4% |
93%/95% |
1%/1% |
Stroke |
4%/3% |
95%/96% |
1%/1% |
Skin Cancer |
10%/6% |
90%/93% |
0%/0% |
Other Cancer |
10%/7% |
90%/93% |
0%/0% |
COPD |
8%/7% |
91%/93% |
1%/0% |
Arthritis |
33%/25% |
66%/75% |
1%/0% |
Mental Health |
34%/37% |
66%/63% |
0%/0% |
Kidney Issues |
4%/3% |
96%/97% |
0%/0% |
Diabetes |
14%/11% |
86%/89% |
0%/0% |
Reviewer comment 8: Dependent variables: It is stated: “Dependent variables in the study were dichotomously coded (0 = absence of morbidity, 1 = presence of morbidity) and included overweight / obese, heart disease, stroke, skin cancer, cancer, COPD, arthritis, mental health condition, kidney disease, and diabetes. In all cases, “do not know”, “refused”, “unknown”, and “missing” responses were coded 0 (not present).”
- Currently it isn’t clear how morbidity status was obtained for this study?
Author response: Our Study Team apologizes for not being clear in this respect. “Morbidity” was a generic term for any of the possible selected dependent variables (Table 3). It was not a variable in itself. To be sure there was no confusion, we added the following.
All dependent variables in the study were dichotomously coded. In all cases, the categories of “Don't Know/Not Sure” and “Not Asked / Missing” were coded with the modal response, as the proportion of these values was 1% or less. For obesity, the modal response was greater than 25% body mass index. For all other dependent variables, the modal response was “No.” For all variables other than obesity, the proportion missing was 1% or less (negligible). For obesity, about 10% of the observations were in the cat-egories “Don't Know/Refused/Missing.” The modal response reflecting overweight / obesity status (greater than 60% of the respondents) is likely to best categorize these in-dividuals. Table 3 shows the unweighted and weighted proportions in the categories of “Yes”, “No”, and “Unknown” for the dependent variables in year 2019.
- Based on what questions/ information from the BRSFF database was it constructed?
Author response: Thank you for pointing this. In response to this, we added Table 2 which should clarify this.
- Regarding the re-coding ““do not know”, “refused”, “unknown”, and “missing” responses were coded 0 (not present)” – First it is recommended to report the numbers; in addition typically those responses are excluded from the analysis. It is not clear way it was re-coded as non- present. Could the authors justify the decision?
Author response: Thank you and yes. Please see Table 3 for these numbers in the Year 2019. Because they are 1% or less for all variables in the study except overweight / obesity, we imputed the modal responses. For overweight / obesity, there were <10% in these categories, so we assigned them the modal response (overweight / obese) as well. We hope this clarifies any confusion in this matter.
Table 3. Year 2019 dependent variable proportions are shown (sample / population estimate).
Question * |
Yes |
No |
Unknown |
Overweight / Obese |
62%/60% |
29%/30% |
9%/10% |
Coronary Heart Disease |
6%/4% |
93%/95% |
1%/1% |
Stroke |
4%/3% |
95%/96% |
1%/1% |
Skin Cancer |
10%/6% |
90%/93% |
0%/0% |
Other Cancer |
10%/7% |
90%/93% |
0%/0% |
COPD |
8%/7% |
91%/93% |
1%/0% |
Arthritis |
33%/25% |
66%/75% |
1%/0% |
Mental Health |
34%/37% |
66%/63% |
0%/0% |
Kidney Issues |
4%/3% |
96%/97% |
0%/0% |
Diabetes |
14%/11% |
86%/89% |
0%/0% |
*May not add to 100% due to rounding
- The authors’ state: “Many of the dependent variables did not have observations dating back to 2003. Table 2 shows the analysis starting year for each of the variables.” - However could the authors please clarify did the information was collected across all 50 states as well as the District of Columbia and 3 U.S. territories
Author response: Thank you for your question. Our Study Team used complete data and now specify that these include the District of Columbia (DC) and the three U.S. territories.
The authors combined data analysis and other potential variables under dependent Which is extremely confusing and hard to follow. It would be recommended that the authors would create two separate subheading. For example first subheading: Study Covariates for example: here included race/ethnicity (white, black and other race/ethnicity), age (18–24, 25–34, 35–44, 45–54, 55–64) etc... In addition, it would recommend excluding the SASS variable code for example “_IMPRACE”. The second subheading: Data Analysis- that include clear description of data analysis that been carried in the study. That isn’t clear.
Author response: Thank you so much for this observation. In an effort to better clarify this area, our Study Team has inserted separate headings as follows:
2.1. Data Source
2.2. Study Sample
2.3. Study Measures
2.3.1. Dependent Variables
2.3.2. Independent Variables
2.4. Methods and Models
2.4.1. Time Series
2.4.2. General Linear Models
2.4.3. Recoding for Covariates
2.5. Data Analysis
Reviewer comment 9: Results
It isn’t clear -How much participants included in the study; how much was in the veterans ? how much was in the non-veterans , how much was in each chronic conditions. The authors should consider present table of weighted percentages for demographics and chronic conditions variables by veteran and non-veteran
Author response: Thank you for your observation. In order to address your concern, our Study Team revamped Table 1 to illustrate population estimates and sample sizes by Veteran status. We are hopeful that this addressed your concerns and made it more understandable to the Reader.
Year |
Non-Vet Population |
Vet Population |
Population Estimate |
Non-Vet Sample |
Vet Sample |
Sample |
2003 |
190,348,049 |
30,003,072 |
220,351,121 |
228,159 |
36,525 |
264,684 |
2004 |
191,637,278 |
29,746,086 |
441,734,485 |
260,982 |
42,840 |
303,822 |
2005 |
194,578,583 |
29,532,523 |
445,494,470 |
305,107 |
51,005 |
356,112 |
2006 |
198,138,945 |
29,118,914 |
451,368,965 |
304,989 |
50,721 |
355,710 |
2007 |
202,498,717 |
27,673,461 |
457,430,037 |
370,990 |
59,922 |
430,912 |
2008 |
205,615,985 |
27,244,684 |
463,032,847 |
358,433 |
56,076 |
414,509 |
2009 |
208,756,506 |
26,249,349 |
467,866,524 |
374,909 |
57,698 |
432,607 |
2010 |
211,037,577 |
26,048,662 |
472,092,094 |
390,643 |
60,432 |
451,075 |
2011 |
212,198,501 |
25,812,791 |
475,097,531 |
441,873 |
64,594 |
506,467 |
2012 |
216,959,427 |
26,098,283 |
481,069,002 |
415,817 |
59,870 |
475,687 |
2013 |
219,968,409 |
26,056,006 |
489,082,125 |
430,268 |
61,505 |
491,773 |
2014 |
220,704,167 |
27,778,365 |
494,506,947 |
402,544 |
62,120 |
464,664 |
2015 |
224,174,518 |
27,172,620 |
499,829,670 |
383,614 |
57,842 |
441,456 |
2016 |
227,144,466 |
27,006,670 |
505,498,274 |
422,384 |
63,919 |
486,303 |
2017 |
229,254,924 |
26,398,281 |
509,804,341 |
392,148 |
57,868 |
450,016 |
2018 |
230,694,063 |
27,379,324 |
513,726,592 |
381,382 |
56,054 |
437,436 |
2019 |
226,740,688 |
25,689,603 |
510,503,678 |
365,038 |
53,230 |
418,268 |
Totals |
3,610,450,803 |
465,008,694 |
7,898,488,703 |
6,229,280 |
952,221 |
7,181,501 |
Reviewer comment 10: Discussion
The author’s state: “Multiple deployments over the last several decades may be a major contributor to increased illness seen after separation from the service [30]. The demand placed upon the U.S. Service member, particularly since the mid to late 1980’s has been illustrated in the number of military operations requiring deployment [20]. Unfortunately, the number of available personnel to support these deployments has not kept pace, resulting in more Service members deploying more often. As depicted in the health outcomes data, this combination of increased demand in military operations placed upon a smaller population may be resulting in a group of at-risk individuals for a variety of debilitating health conditions [1]. Additional stressors may exist in the occupational requirement for all Service members to meet stringent height and weight standards, and physical fitness standards, both of which are evaluated every six months [17]. Perhaps the combination of each of these stressors ultimately contribute to the stark differences in health outcomes between the Veteran populations compared to non-Veteran populations [1-3] As depicted in the health outcomes data, this combination of increased demand in military operations placed upon a smaller population may be resulting in a group of at-risk individuals for a variety of debilitating health conditions [1]. Additional stressors may exist in the occupational requirement for all Service members to meet stringent height and weight standards, and physical fitness standards, both of which are evaluated every six months [17].”
Yet the study didn’t examine any of the variables [Information on deployment or height and weight standards, and physical fitness standards] the authors cannot provide data. This section should be remove.
Author response: Thank you for your comments and although the Study Team fully appreciates the Reviewer’s suggestion of removal of this section, it is our collective belief that the revamped tables and analyses strengthen the argument the Service members throughout this period have faced an increasing number of stressors which include maintaining a high level of physical fitness, maintaining strict height and weight standards, and an increase in operational deployments which inherently include potential family separation. Removal of this section would greatly weaken this argument.
Reviewer 2 Report
The collective health of those who served in the U.S. military exhibits a number of health conditions that paint a concerning picture of morbidity among the U.S. Veteran population. The combination of increased demand in military operations placed upon a smaller population may be resulting in a group of at-risk individuals for a variety of debilitating health conditions. So the aim of the study was to explore health outcomes for U.S. Veterans compared to Non-Veterans from 2003 to 2019.
The paper has a clear friendly structure (Introduction, Materials and Methods, Results, Discussion and Conclusions). The subject is interesting and useful and raises the worrying issue of the collective poor health of the U.S. Veterans. The introduction is quite informative and the subject is thoroughly explained. The manuscript stands good in- depth analysis of the results and transparent and sufficient discussion containing the limitation section. The text is complemented by three tables, six figures and enriched with 34 adequate references.
However there are a few issues that could be supplemented:
- Materials and Methods section: could you please explain on what basis did you decide:
- as regards ‘marital status’ - the small proportion of “refused to answer” or left blank were assigned to the modal category (unweighted), 1 = married ?;
- as regards ‘education status’ - the small proportion of “refused to answer” or left blank were assigned to the modal category (unweighted), 6 =college 4 or more years?;
- as regards ‘income’ the small proportion of refusals and ‘don`t know’ were assigned to the modal category (unweighted), 1= employed for wages?
- Discussion: In my opinion it would be useful to focus more deeply on the important question: why rates of self-reported mental health disorders were not higher among Veterans than among non-Veterans, since it is well known that military service can often result in posttraumatic stress disorders, e.g. depression and other mental health disorders among veterans?
Round 2
Reviewer 1 Report
Dear Editor,
Thank you for inviting me to review the revised manuscript. The authors have provided a comprehensive reply to the comments raised in the previous round of review, which were noted. The authors also presented a greatly improved manuscript. After reading the revised manuscript, there are additional suggestions in response to the new content. I believe addressing these will enhance the contribution of the paper to the literature. The questions/concerns are outlined below.
Major comments
- While the authors aimed to “ compare Veteran and non-Veteran health outcomes, yearly age-adjusted, population estimates from 2003 (or earliest year available) to 2019 were used for Veteran / non-Veteran” [On p4 ln 121-122 ]. Given the change in BRFSS sampling methods as stated by the authors on p4 ln 121-122: “Because of the change in BRFSS sampling methods beginning in 2011, the CDC warns that “The BRFSS 2011 data should be considered a baseline year for data analysis and is not directly comparable to previous years of BRFSS data because of the changes in weighting methodology and the addition of the cell phone sampling frame [24].” Based on BRSFF recommendation the authors should strongly consider excluding 2003- 2010 form the paper. I would like to refer the authors to Table 4. Starting years for analysis. Five dependent variables (Skin Cancer, Cancer, COPD, Arthritis, Kidney Disease) BRSFF only started data collection 2010-2011. Finally, I would like also to refer the authors to the results section, table 5 and 6 narrative discuss only 2011-2019. There is no results discussion 2003-2010
- Throughout the discussion, the authors argue that their findings of higher health morbidities among Veterans, is due to multiple deployments and/or additional stressors that may exist in the occupational requirement for all Service members. Yet this claim is not supported by the current findings. To show disproportionate impact the authors would need to have demonstrated that any worsening in health morbidities across the time period was more marked for the Veterans group, either by stratifying their analysis or by generating a separate measure for multiple deployments and/or additional stressors additional occupational requirement.
- On the other hand, I feel there is lack of explanation of demographic, socio-economic, geographic and how it may explain the study results. I would dedicate more attention and discuss a bit further the study results in relation to SES factors and add the relevant literature.
- They found that self-reported mental health disorders were not higher for Veterans when comparing to non-Veterans. There is some discussion regarding negative consequences of reporting mental health issues in the military within the manuscript, but this finding appears to specifically highlight a potential ongoing issue in accessing services and should be discussed [screening, assessment vs treatments];
Minor comments
1. In Table 5: To improve readability, I would advise making the table title clearer and to match the results [missing the age-adjusted comparisons]. In addition, through the paper the depended variable is mental health, while in Table 5: it is depression.
Author Response
Thank you so much for the time it took for you to review our article. And although we are most grateful for your comments, we must emphasize that our Study Team did not combine the data other than from 2011 through 2019 for our analysis. Indeed, we use each year as an independent analysis prior to 2011. The reason for that is based exactly upon the following quote which I have in the paper:
While the authors aimed to “ compare Veteran and non-Veteran health outcomes, yearly age-adjusted, population estimates from 2003 (or earliest year available) to 2019 were used for Veteran / non-Veteran” [On p4 ln 121-122 ]. Given the change in BRFSS sampling methods as stated by the authors on p4 ln 121-122: “Because of the change in BRFSS sampling methods beginning in 2011, the CDC warns that “The BRFSS 2011 data should be considered a baseline year for data analysis and is not directly comparable to previous years of BRFSS data because of the changes in weighting methodology and the addition of the cell phone sampling frame [24].” Based on BRSFF recommendation the authors should strongly consider excluding 2003- 2010 form the paper. I would like to refer the authors to Table 4. Starting years for analysis. Five dependent variables (Skin Cancer, Cancer, COPD, Arthritis, Kidney Disease) BRSFF only started data collection 2010-2011. Finally, I would like also to refer the authors to the results section, table 5 and 6 narrative discuss only 2011-2019. There is no results discussion 2003-2010
The best estimate for the years PRIOR to 2011 still comes from the BRFSS. We just cannot COMBINE prior to 2011. However, the data are still the best estimates when looked at independently. The difference is the following: for some analysis, we could put 2011-2019 into one big table and apply the weights. They would make sense, as you are estimating the populations for 2011-2019 (sum of the population). However, if you added 2010, then the weights would NOT make sense, because the weighting was done differently. We are hopeful that this explanation fully addresses your concerns. Thank you again.
Comment 2: Throughout the discussion, the authors argue that their findings of higher health morbidities among Veterans, is due to multiple deployments and/or additional stressors that may exist in the occupational requirement for all Service members. Yet this claim is not supported by the current findings. To show disproportionate impact the authors would need to have demonstrated that any worsening in health morbidities across the time period was more marked for the Veterans group, either by stratifying their analysis or by generating a separate measure for multiple deployments and/or additional stressors additional occupational requirement.
Author response: Thank you for this recommendation. We have highlighted this as being an area for future research to help explain the worsening health among the Veteran population.
Comment 3: On the other hand, I feel there is lack of explanation of demographic, socio-economic, geographic and how it may explain the study results. I would dedicate more attention and discuss a bit further the study results in relation to SES factors and add the relevant literature. They found that self-reported mental health disorders were not higher for Veterans when comparing to non-Veterans. There is some discussion regarding negative consequences of reporting mental health issues in the military within the manuscript, but this finding appears to specifically highlight a potential ongoing issue in accessing services and should be discussed [screening, assessment vs treatments].
Author response: Thank you for this recommendation. Our Study Team has highlighted this with the following addition to the manuscript, “Additionally, Veteran services to address the need for mental health services such as screening, assessment, and treatments continue to challenge the Veteran’s Administration and the population they serve.”
Reviewer 2 Report
Thank you for the corrections and additions. I am very happy with them.
I have no more comments.
Author Response
Author response: Thank you so much and our Study Team is pleased with your response.
Reviewer 3 Report
Excellent updates to methods and clarification of limitations and recommendations for future research.
Author Response

(The authors gave the same response as above.)

Round 3
Reviewer 1 Report
The authors have clarified several of the questions I raised in my previous review. I respect the authors’ patience and professionalism in dealing with what I can only assume is a rather harsh review experience.
Yet I have some major concerns with the study discussion that prevent me to endorse its acceptance at the present stage. The detail comments are given below.
In respect to my comment #1 (methods and results sections): The revised methods match the manuscript study aims and the BRSFF datasets data collections and variables.
In respect to my comment #2-4: The discussion do not match the study results. Both the introduction and discussion address stressors associated with multiple deployments. The authors now acknowledge in their responses to comment #1 and #2: “This study sought to utilize the BRFSS data to provide a snap-shot or “glimpse” into a possible correlation between the stressors associated with multiple deployments, and poor health outcomes”.
- However, this was not stated study aims; “In this study, we investigate morbidities between Veteran and non-Veterans and over time with the belief that Veterans are likely to experience more morbidities due to the difficulties and exposures of military service. Further, we use data from 2003-2010 and aggregated data from 2011 through 2019, which includes the newest release of the Centers for Disease Control and Prevention (CDC) Behavioral Risk Factor Surveillance System (BRFSS) [21-23] dataset, to look at demographic, socio-economic, geographic, time and veteran status variables that may assist in explaining any differences”. [On p4 ln 113-119].
- As I stated in my previous reviews the BRSFF dataset, do not collect any information regarding military service factors [deployments, length of time in service, branch of service, and/or combat exposure]. The study results only support been veteran or prior military service with
- The discussion should address the demographic socioeconomic and geographic variables that were found in the results; this would help identify potential risk factors, which is the goal of BRSFF and would help prompt federal agencies or Veteran advocacy groups to better inform policy decisions in order to meet the health needs of Veterans as the authors stated.
- To help and increases the study findings and rigors of the discussion the authors should also include prior studies that utilized the BRSFF data to examine health outcomes among U.S. Military and veteran. Here is some examples:
- Walker LE, Poltavskiy E, Janak JC, Beyer CA, Stewart IJ, Howard JT. US Military Service and Racial/Ethnic Differences in Cardiovascular Disease: An Analysis of the 2011-2016 Behavioral Risk Factor Surveillance System. Ethn Dis. 2019 Jul 18;29(3):451-462. doi: 10.18865/ed.29.3.451. PMID: 31367165; PMCID: PMC6645722.
- Hoerster KD, Lehavot K, Simpson T, McFall M, Reiber G, Nelson KM. Health and health behavior differences: U.S. Military, veteran, and civilian men. Am J Prev Med. 2012 Nov;43(5):483-9. doi: 10.1016/j.amepre.2012.07.029. PMID: 23079170.
- Lehavot K, Hoerster KD, Nelson KM, Jakupcak M, Simpson TL. Health indicators for military, veteran, and civilian women. Am J Prev Med. 2012 May;42(5):473-80. doi: 10.1016/j.amepre.2012.01.006. PMID: 22516487
Author Response
Thank you again for granting us the opportunity to respond to you regarding our draft manuscript titled, “Exploring Health Outcomes for U.S. Veterans Compared to Non-Veterans from 2003 to 2019.“ Our Study Team is most appreciative for your time, effort and professionalism displayed by you as your comments are most insightful and substantive in nature. Please permit me to provide the following responses to you.
Reviewers’ Comments to the Authors Followed by Responses:
Comment 1: In respect to my comment #1 (methods and results sections): The revised methods match the manuscript study aims and the BRSFF datasets data collections and variables.
Author response: Thank you for acknowledging that the revised methods match the manuscript study aims and the BRFSS datasets, data collections and variables.
Comment 2: In respect to my comment #2-4: The discussion do not match the study results. Both the introduction and discussion address stressors associated with multiple deployments. The authors now acknowledge in their responses to comment #1 and #2: “This study sought to utilize the BRFSS data to provide a snap-shot or “glimpse” into a possible correlation between the stressors associated with multiple deployments, and poor health outcomes”. However, this was not stated study aims; “In this study, we investigate morbidities between Veteran and non-Veterans and over time with the belief that Veterans are likely to experience more morbidities due to the difficulties and exposures of military service. Further, we use data from 2003-2010 and aggregated data from 2011 through 2019, which includes the newest release of the Centers for Disease Control and Prevention (CDC) Behavioral Risk Factor Surveillance System (BRFSS) [21-23] dataset, to look at demographic, socio-economic, geographic, time and veteran status variables that may assist in explaining any differences”. [On p4 ln 113-119].
Author response: Thank you for this valuable comment and recommendation. Our Study Team has revised the study aim accordingly. Now, it reads “The specific aim of this study is to provide a snapshot or a “glimpse” into potential relationships between the stressors associated with multiple deployments, and poor health outcomes among US Veterans. We specifically investigate morbidities between Veteran and non-Veterans and over time with the belief that Veterans are likely to experience more morbidities due to the difficulties and exposures of military service. Further, we use data from 2003-2010 and aggregated data from 2011 through 2019, which include the newest release of the Centers for Disease Control and Prevention (CDC) Behavioral Risk Factor Surveillance System (BRFSS) [21-23] dataset, to look at demographic, socio-economic, geographic, time and Veteran status variables that may assist in explaining any differences.”
Comment 3: As I stated in my previous reviews the BRSFF dataset, do not collect any information regarding military service factors [deployments, length of time in service, branch of service, and/or combat exposure]. The study results only support been veteran or prior military service with . . . .
Author response: Thank you for your comment as our Study Team agrees with the Reviewer and we added a section in the “Discussion” lack of variables such as number of deployments, length of time in service, branch of service, combat exposure, length of time between service termination and survey year as limitations to this study.
“ Unfortunately, although the BRFSS dataset does illustrate those who identify themselves as ‘Veterans,’ it lacks the desired level of detail to fully define an association between multiple deployments and poor health outcomes.”
We also included the lack of the variables above as limitations to this study.
“Furthermore, due to data limitation, this study was not able to include variables associated with military services, such as military rank, the number of deployments, the length of time in service, the branch of service, the type of combat or exposure, the length of time between deployment and survey year, the number of injuries, the types of injuries, and the combat geographic locations. Future studies that include these variables are called upon.”
3. The discussion should address the demographic socioeconomic and geographic variables that were found in the results; this would help identify potential risk factors, which is the goal of BRSFF and would help prompt federal agencies or Veteran advocacy groups to better inform policy decisions in order to meet the health needs of Veterans as the authors stated
Author response: “ As such some specific health screening, prevention, and health care services should be provided to specific population based on our findings. Regarding the demographic, socioeconomic and geographic variables, our findings suggest that odds of being overweight, being diagnosed with a heart disease, skin cancer, or other cancers, COPD, arthritis, kidney disease and diabetes, as well as the odds of having a stroke increase with age. Walker et al., 2019 also found that weight and age were positively associated with the odds of having a heart disease. In addition, being a Caucasian is associated with increased odds of having a heart disease, skin cancer, other cancers, arthritis, COPD, and mental health issues, but associated with lower odds of being overweight, having a stroke, or a kidney disease compared with being a non-Caucasian. Hispanics have a better health status than their non-Hispanic counterparts. While Hispanics have higher odds of being overweight, having a skin cancer, and a kidney disease, they have lower odds of having a heart disease, stroke, other cancers, COPD, arthritis, mental health issue compared with non-Hispanics. Our finding that Hispanics are less likely to have heart disease is supported by previous study on veterans (Walker et al., 2019)
Also compared to being a female, being a male is associated with higher odds of being overweight, as well as having a heart disease, stroke, skin cancer and diabetes, but it is associated with lower odds of having other cancers, COPD, arthritis, mental health issues and kidney disease. With regards to marital status and annual income, married individuals and those with annual income of ≥ $75,000 have a better health status than singles and those who have lower income. They are less likely to have heart disease stroke, other cancers, COPD, arthritis, mental health issues, kidney disease and diabetes, but more likely to be overweight and having skin cancer, compared with their single and lower income counterparts. In the same vein, being college graduates and employed for wages are associated with a better health status compared with those who are not collage graduate and are not employed for wages. Being a college graduate was associated with lower odds of all having these health issues except for skin cancer and being employed for wage was only associated with being overweight. Our findings regarding the associations between gender and heart disease as well as income and heart disease are supported by (Walker et al., 2019)
With respect to geographic location, individuals who live in the East North Central, Middle Atlantic, Mountain, Pacific, South Atlantic, West South Central regions as well as in the Territories of U.S. are more likely to have skin cancer, while individuals who reside in the West North Central region are more likely to have a heart disease.”
Comment 4: To help and increases the study findings and rigors of the discussion the authors should also include prior studies that utilized the BRSFF data to examine health outcomes among U.S. Military and veteran. Here is some examples:
- Walker LE, Poltavskiy E, Janak JC, Beyer CA, Stewart IJ, Howard JT. US Military Service and Racial/Ethnic Differences in Cardiovascular Disease: An Analysis of the 2011-2016 Behavioral Risk Factor Surveillance System. Ethn Dis. 2019 Jul 18;29(3):451-462. doi: 10.18865/ed.29.3.451. PMID: 31367165; PMCID: PMC6645722.
- Hoerster KD, Lehavot K, Simpson T, McFall M, Reiber G, Nelson KM. Health and health behavior differences: U.S. Military, veteran, and civilian men. Am J Prev Med. 2012 Nov;43(5):483-9. doi: 10.1016/j.amepre.2012.07.029. PMID: 23079170.
- Lehavot K, Hoerster KD, Nelson KM, Jakupcak M, Simpson TL. Health indicators for military, veteran, and civilian women. Am J Prev Med. 2012 May;42(5):473-80. doi: 10.1016/j.amepre.2012.01.006. PMID: 22516487”
Author response: Thank you for your suggestion. We compared our findings with the findings from Walker et al (2019) in the Discussion section. However, our study was not comparable with those of Hoerster et al, 2012 and Lehavot at al. 2012 because both studies conducted gender-specific studies while our study sample included both men and women.
Again, thank you so much for the time it took to review our manuscript. I welcome any additional questions or comments on our revised manuscript. Thank you.
This manuscript is a resubmission of an earlier submission. The following is a list of the peer review reports and author responses from that submission.
Round 1
Reviewer 1 Report
Dear Editor,
Thank you for inviting me to review this manuscript. The aim of this study was to better understand how the U.S Veterans health outcomes compared to non-Veterans. This study utilized the Behavioral Risk Factor Surveillance System (BRFSS) dataset from the period 2003 through 2019, to examine 10 different self-reported morbidities including overweight / obesity, heart disease, stroke, skin cancer, cancer, COPD, arthritis, mental health, kidney disease, and diabetes. I believe that the topic is very interesting with scientific interest and that the authors put effort in drafting this work. However, the current version of the manuscript is let down by its written presentation, the results are not well presented and important details in the methods are lacking or not clear. Nevertheless, I offer comments and suggestions for the author’s consideration.
Introduction
The authors will need to spend time with the Introduction of their manuscript to describe the concept of “multiple chronic conditions” and the possible associations with behavioral risk factors. To help I would strongly recommend to look at the following references that also utilized the BRSFF data.
- Adams, M. (2019). The impact of key modifiable risk factors on leading chronic conditions. Preventive Medicine., 120, 113–118. https://doi.org/10.1016/j.ypmed.2019.01.006
- Adams, M. (2017). Linear association between number of modifiable risk factors and multiple chronic conditions: Results from the Behavioral Risk Factor Surveillance System. Preventive Medicine., 105, 169–175. https://doi.org/10.1016/j.ypmed.2017.09.013
- Stone C. L. (2020). A population-based measure of chronic disease severity for health planning and evaluation in the United States. AIMS public health, 7(1), 44–65. https://doi.org/10.3934/publichealth.2020006
Methods
General comment: The methods section is a bit confusing and hard to follow. To help the reader it would be recommended to re-structure the section. Recommendation to re-structure: Data source ->Study Sample-> Institutional Review Board (IRB) -> Study Measures: Dependent Variable, Independent Variables, Study covariates-> Data Analysis. In addition, some additional details and clarification of the methods used would be helpful.
- Data: To demonstrate the strength of the BRSFF data. It would be recommended to expend the data sub-section. For example, how is the data collected? How many interviews per year are collected? In addition, there should be explanation regarding the BRSFF questionnaire protocol.
- The authors have stated: “The BRFSS is a continuous, surveillance system that collects information from each state in the U.S. about modifiable risk factors for chronic diseases and other leading causes of death [22]”. However, the BRFSS now collects data in all 50 states as well as the District of Columbia and 3 U.S. territories.
- The authors have also stated: “Each year was used to estimate the population point estimate for the specific time and morbidity. These data are considered by the CDC to be the nation’s best source for health-related survey data, and the 2019 version includes 418,268 observations. Applying complex weighting to these observations results in a picture of the entire nation’s health self-reported health [23].” First, I would recommend that this section should be moved under sampling. Second, it is not clear what the authors mean “complex weighting”. In 2011, BRFSS began using a weighting method known as iterative proportional fitting, or raking. Could the authors please clarify the sampling section?
- Institutional Review Board- Currently the manuscript is missing IRB statement the authors should address this.
- Software: I would recommend that this section should be moved under Data Analysis.
- Sample Size: It isn’t clear how the sample was construct and the authors should expend the section
- The authors have stated: “Table 1 shows the sample size for each of the years in question” - Could the authors clarify what table 1 present? Is it weighted percentage or unweighted sample size? Is the sample include both veteran and non-veteran?
- The authors also stated “ Complex weights were applied to estimate the entire population, resulting in fractional values”. – Yet it is not clear how “complex weight were applied”. There should be clear explanation regarding how the 2003-2019 survey data were combined, weighted percentages was calculated what for what variables and which groups
- The independent variable: It is stated: “The primary variables of interest were the year of the estimate (2003 through 2018), and Veteran status, coded {0 = non-Veteran, 1= Veteran} with blank, refused, and do not know coded as non-Veteran”
- Currently it isn’t clear how veteran status variable was obtained for this study?
- Based on what question/ information from the BRSFF data base was it constructed?
- Regarding the re-coding “blank, refused, and do not know coded as non-Veteran”
- Blank are typically state that were not asked the questions, refused and don’t know are consider missing data. First, it is recommended to report the numbers; in addition typically those responses are excluded from the analysis. It is not clear way it was re-coded as non-Veteran. Could the authors justify the decision?
- Dependent variables: It is stated: “Dependent variables in the study were dichotomously coded (0 = absence of morbidity, 1 = presence of morbidity) and included overweight / obese, heart disease, stroke, skin cancer, cancer, COPD, arthritis, mental health condition, kidney disease, and diabetes. In all cases, “do not know”, “refused”, “unknown”, and “missing” responses were coded 0 (not present).”
- Currently it isn’t clear how morbidity status was obtained for this study?
- Based on what questions/ information from the BRSFF database was it constructed?
- Regarding the re-coding ““do not know”, “refused”, “unknown”, and “missing” responses were coded 0 (not present)” – First it is recommended to report the numbers; in addition typically those responses are excluded from the analysis. It is not clear way it was re-coded as non- present. Could the authors justify the decision?
- The authors’ state: “Many of the dependent variables did not have observations dating back to 2003. Table 2 shows the analysis starting year for each of the variables.” - However could the authors please clarify did the information was collected across all 50 states as well as the District of Columbia and 3 U.S. territories
- The authors combined data analysis and other potential variables under dependent Which is extremely confusing and hard to follow. It would be recommended that the authors would create two separate subheading. For example first subheading: Study Covariates for example: here included race/ethnicity (white, black and other race/ethnicity), age (18–24, 25–34, 35–44, 45–54, 55–64) etc... In addition, it would recommend excluding the SASS variable code for example “_IMPRACE”. The second subheading: Data Analysis- that include clear description of data analysis that been carried in the study. That isn’t clear.
Results
It isn’t clear -How much participants included in the study; how much was in the veterans ? how much was in the non-veterans , how much was in each chronic conditions. The authors should consider present table of weighted percentages for demographics and chronic conditions variables by veteran and non-veteran
Discussion
The author’s state: “Multiple deployments over the last several decades may be a major contributor to increased illness seen after separation from the service [30]. The demand placed upon the U.S. Service member, particularly since the mid to late 1980’s has been illustrated in the number of military operations requiring deployment [20]. Unfortunately, the number of available personnel to support these deployments has not kept pace, resulting in more Service members deploying more often. As depicted in the health outcomes data, this combination of increased demand in military operations placed upon a smaller population may be resulting in a group of at-risk individuals for a variety of debilitating health conditions [1]. Additional stressors may exist in the occupational requirement for all Service members to meet stringent height and weight standards, and physical fitness standards, both of which are evaluated every six months [17]. Perhaps the combination of each of these stressors ultimately contribute to the stark differences in health outcomes between the Veteran populations compared to non-Veteran populations [1-3] As depicted in the health outcomes data, this combination of increased demand in military operations placed upon a smaller population may be resulting in a group of at-risk individuals for a variety of debilitating health conditions [1]. Additional stressors may exist in the occupational requirement for all Service members to meet stringent height and weight standards, and physical fitness standards, both of which are evaluated every six months [17].”
Yet the study didn’t examine any of the variables [Information on deployment or height and weight standards, and physical fitness standards] the authors cannot provide data. This section should be remove.
Author Response
Thank you for giving us the opportunity to submit a revised draft of the manuscript, “Exploring Health Outcomes for U.S. Veterans Compared to Non-Veterans from 2003 to 2019“ for publication in Healthcare. We appreciate the time and effort that you and your reviewers dedicated to providing feedback on our manuscript and are grateful for the insightful comments on the valuable improvements to our paper. We have incorporated the recommendations made by the reviewers. These changes are highlighted within the manuscript for your ease in reading. Please see below (in blue text), for a point-by-point response to the reviewers’ comments and concerns. All page numbers refer to the revised manuscript file with tracked changes.
Reviewer comment 1: Thank you for inviting me to review this manuscript. The aim of this study was to better understand how the U.S Veterans health outcomes compared to non-Veterans. This study utilized the Behavioral Risk Factor Surveillance System (BRFSS) dataset from the period 2003 through 2019, to examine 10 different self-reported morbidities including overweight / obesity, heart disease, stroke, skin cancer, cancer, COPD, arthritis, mental health, kidney disease, and diabetes. I believe that the topic is very interesting with scientific interest and that the authors put effort in drafting this work.
Author response: Thank you for your kinds comments. Yes, our Study Team saw a need to address this very important topic for a population that has sacrificed so much.
However, the current version of the manuscript is let down by its written presentation, the results are not well presented and important details in the methods are lacking or not clear. Nevertheless, I offer comments and suggestions for the author’s consideration.
Author response: We truly appreciate your valuable assistance in improving this important manuscript. We have embraced your concerns and recommended modifications and have added a new online reference that provides all of our variable encoding / recoding / feature engineering as well as all of our analysis for replication.
“To compare Veteran and non-Veteran health outcomes, yearly age-adjusted, population estimates from 2003 (or earliest year available) to 2019, were used for Veteran / non-Veteran. The entirety of the analysis is available for review at the following location https://rpubs.com/R-Minator/BRFSS.”
Reviewer comment 2: Introduction
The authors will need to spend time with the Introduction of their manuscript to describe the concept of “multiple chronic conditions” and the possible associations with behavioral risk factors. To help I would strongly recommend to look at the following references that also utilized the BRSFF data.
- Adams, M. (2019). The impact of key modifiable risk factors on leading chronic conditions. Preventive Medicine., 120, 113–118. https://doi.org/10.1016/j.ypmed.2019.01.006
- Adams, M. (2017). Linear association between number of modifiable risk factors and multiple chronic conditions: Results from the Behavioral Risk Factor Surveillance System. Preventive Medicine., 105, 169–175. https://doi.org/10.1016/j.ypmed.2017.09.013
- Stone C. L. (2020). A population-based measure of chronic disease severity for health planning and evaluation in the United States. AIMS public health, 7(1), 44–65. https://doi.org/10.3934/publichealth.2020006
Author response: Thank you for your insightful recommendations. We have added these recommended references.
Reviewer comment 3: Methods
General comment: The methods section is a bit confusing and hard to follow. To help the reader it would be recommended to re-structure the section. Recommendation to re-structure: Data source ->Study Sample-> Institutional Review Board (IRB) -> Study Measures: Dependent Variable, Independent Variables, Study covariates-> Data Analysis. In addition, some additional details and clarification of the methods used would be helpful.
Author response: Thank you for your comment and recommendations. Based on all of your comments, we have restructured as follows.
- Materials and Methods
2.1. Data Source
2.2. Study Sample
2.3. Study Measures
2.3.1. Dependent Variables
2.3.2. Independent Variable
2.3.3. Covariates
2.3.3.1. Demographic Variables
2.3.3.2. Socio-Economic Variables
2.3.3.3. Geographic Variables
2.3.3.4. Time Variable
2.4. Methods & Models
2.4.1. Descriptive Models
2.4.2. General Linear Models
2.4.3. Data Analysis
- Results
3.1. Descriptive Statistics
3.2. General Linear Models
- Data: To demonstrate the strength of the BRSFF data. It would be recommended to expand the data sub-section. For example, how is the data collected? How many interviews per year are collected? In addition, there should be explanation regarding the BRSFF questionnaire protocol.
Author response: Thank you very much for your comments regarding the structure and information needed for the methods section. Our Study Team re-structured this section based on your recommendation to help with the flow of the information. Regarding the data source, we included additional information regarding the methods of data collections for the BRFSS, where it is collected and also added in an additional reference to the CDC website where a robust methodology on the BRFSS can be readily found. We also provided an updated Table 1 that illustrates the sheer size of the samples.
2.2. Study Sample
For this study, BRFSS data from 2003 to 2019 (last available year) were acquired from the CDC [21]. Each year was used to estimate the population point estimate for the specific time and morbidity. The BRFSS data are considered by the CDC to be the nation’s best source for health-related survey data, and the 2019 version includes 418,268 observations. Applying stratum and individual weights to these observations results in a picture of the entire nation’s health self-reported health [23]. En toto, this study included 7.181 million sample observations.
Table 1 shows the population estimates and sample sizes for each of the years in question by Veteran / Non-Veteran status. Sampling weights based on location and socio-demographic information were applied to estimate the entire population, resulting in fractional values for population estimates. Prior to 2011, samples were weighted by stratum (area code / prefix combinations) times the inverse number of telephones in the household times the number of adults in the household times the number of people in an age-by-sex or age-by-race/ethnicity-by-sex category in the population of a region or a state divided by the sum of the preceding weights for the respondents in the same age-by-sex or age-by-race/ethnicity-by-sex category (adjusting for noncoverage and nonresponse and forces the sum of the weighted frequencies to equal population estimates for the region or state.) Two separate variable weight fields. This weighting was applied through the use of separate fields for stratum and weights. From 2011 onwards, raking replaced the post-stratification process. Raking allows for incorporation of cellular telephone survey data and allows for inclusion of other demographic considerations to improve socio-demographic matching by stratum.
Table 1. Population estimates and sample sizes by year and veteran status
Year |
Non-Vet Population |
Vet Population |
Population Estimate |
Non-Vet Sample |
Vet Sample |
Sample |
2003 |
190,348,049 |
30,003,072 |
220,351,121 |
228,159 |
36,525 |
264,684 |
2004 |
191,637,278 |
29,746,086 |
441,734,485 |
260,982 |
42,840 |
303,822 |
2005 |
194,578,583 |
29,532,523 |
445,494,470 |
305,107 |
51,005 |
356,112 |
2006 |
198,138,945 |
29,118,914 |
451,368,965 |
304,989 |
50,721 |
355,710 |
2007 |
202,498,717 |
27,673,461 |
457,430,037 |
370,990 |
59,922 |
430,912 |
2008 |
205,615,985 |
27,244,684 |
463,032,847 |
358,433 |
56,076 |
414,509 |
2009 |
208,756,506 |
26,249,349 |
467,866,524 |
374,909 |
57,698 |
432,607 |
2010 |
211,037,577 |
26,048,662 |
472,092,094 |
390,643 |
60,432 |
451,075 |
2011 |
212,198,501 |
25,812,791 |
475,097,531 |
441,873 |
64,594 |
506,467 |
2012 |
216,959,427 |
26,098,283 |
481,069,002 |
415,817 |
59,870 |
475,687 |
2013 |
219,968,409 |
26,056,006 |
489,082,125 |
430,268 |
61,505 |
491,773 |
2014 |
220,704,167 |
27,778,365 |
494,506,947 |
402,544 |
62,120 |
464,664 |
2015 |
224,174,518 |
27,172,620 |
499,829,670 |
383,614 |
57,842 |
441,456 |
2016 |
227,144,466 |
27,006,670 |
505,498,274 |
422,384 |
63,919 |
486,303 |
2017 |
229,254,924 |
26,398,281 |
509,804,341 |
392,148 |
57,868 |
450,016 |
2018 |
230,694,063 |
27,379,324 |
513,726,592 |
381,382 |
56,054 |
437,436 |
2019 |
226,740,688 |
25,689,603 |
510,503,678 |
365,038 |
53,230 |
418,268 |
Totals |
3,610,450,803 |
465,008,694 |
7,898,488,703 |
6,229,280 |
952,221 |
7,181,501 |
- The authors have stated: “The BRFSS is a continuous, surveillance system that collects information from each state in the U.S. about modifiable risk factors for chronic diseases and other leading causes of death [22]”.However, the BRFSS now collects data in all 50 states as well as the District of Columbia and 3 U.S. territories.
Author response: Thank you as you are absolutely correct. Based on your recommendation, we corrected this discussion and actually use a factor-level indicating “Territory” in our analysis.
The BRFSS is a continuous health survey system that collects information from each state in the U.S., three U.S. territories and the District of Columbia about modifiable risk factors for chronic diseases and other leading causes of death [22].
- The authors have also stated: “Each year was used to estimate the population point estimate for the specific time and morbidity. These data are considered by the CDC to be the nation’s best source for health-related survey data, and the 2019 version includes 418,268 observations. Applying complex weighting to these observations results in a picture of the entire nation’s health self-reported health [23].”First, I would recommend that this section should be moved under sampling. Second, it is not clear what the authors mean “complex weighting”. In 2011, BRFSS began using a weighting method known as iterative
Author response: Thank you for your recommendation. Based on your recommendation, we now have a separate section for study sample discussion (see above) and discuss the raking method from 2011 as well in 2.1.4.
From 2011 onwards, raking replaced the post-stratification process. Raking allows for incorporation of cellular telephone survey data and allows for inclusion of other demo-graphic considerations to improve socio-demographic matching by stratum.
Reviewer comment 4: Institutional Review Board- Currently the manuscript is missing IRB statement the authors should address this.
Author response: Thank you for calling this to our attention. BRFSS data is a publicly available, anonymous data set that does not require IRB review or an exempt status for its use. The data is posted to the CDC’s website. The following statement was added into the paper for clarity.
BRFSS data is a publicly available, anonymous data set that does not require institutional review board or an exempt status for its use.
Reviewer comment 5: Software: I would recommend that this section should be moved under Data Analysis.
Author response: Our Study Team fully appreciates this suggestion and we have moved this section under the title of Data Analysis. Thank you.
Reviewer comment 6: Sample Size: It isn’t clear how the sample was construct and the authors should expend the section
- The authors have stated: “Table 1 shows the sample size for each of the years in question” - Could the authors clarify what table 1 present? Is it weighted percentage or unweighted sample size? Is the sample include both veteran and non-veteran?
Author response: Table 1 was revamped to show the sample sizes and population estimates.
Table 1 shows the population estimates and sample sizes for each of the years in question by Veteran / Non-Veteran status. Sampling weights based on location and socio-demographic information were applied to estimate the entire population, resulting in fractional values for population estimates. Prior to 2011, samples were weighted by stratum (area code / prefix combinations) times the inverse number of telephones in the household times the number of adults in the household times the number of people in an age-by-sex or age-by-race/ethnicity-by-sex category in the population of a region or a state divided by the sum of the preceding weights for the respondents in the same age-by-sex or age-by-race/ethnicity-by-sex category (adjusting for noncoverage and nonresponse and forces the sum of the weighted frequencies to equal population estimates for the region or state.) Two separate variable weight fields. This weighting was applied through the use of separate fields for stratum and weights. From 2011 onwards, raking replaced the post-stratification process. Raking allows for incorporation of cellular telephone survey data and allows for inclusion of other demographic considerations to improve socio-demographic matching by stratum.
Table 1. Population estimates and sample sizes by year and veteran status
Year |
Non-Vet Population |
Vet Population |
Population Estimate |
Non-Vet Sample |
Vet Sample |
Sample |
2003 |
190,348,049 |
30,003,072 |
220,351,121 |
228,159 |
36,525 |
264,684 |
2004 |
191,637,278 |
29,746,086 |
441,734,485 |
260,982 |
42,840 |
303,822 |
2005 |
194,578,583 |
29,532,523 |
445,494,470 |
305,107 |
51,005 |
356,112 |
2006 |
198,138,945 |
29,118,914 |
451,368,965 |
304,989 |
50,721 |
355,710 |
2007 |
202,498,717 |
27,673,461 |
457,430,037 |
370,990 |
59,922 |
430,912 |
2008 |
205,615,985 |
27,244,684 |
463,032,847 |
358,433 |
56,076 |
414,509 |
2009 |
208,756,506 |
26,249,349 |
467,866,524 |
374,909 |
57,698 |
432,607 |
2010 |
211,037,577 |
26,048,662 |
472,092,094 |
390,643 |
60,432 |
451,075 |
2011 |
212,198,501 |
25,812,791 |
475,097,531 |
441,873 |
64,594 |
506,467 |
2012 |
216,959,427 |
26,098,283 |
481,069,002 |
415,817 |
59,870 |
475,687 |
2013 |
219,968,409 |
26,056,006 |
489,082,125 |
430,268 |
61,505 |
491,773 |
2014 |
220,704,167 |
27,778,365 |
494,506,947 |
402,544 |
62,120 |
464,664 |
2015 |
224,174,518 |
27,172,620 |
499,829,670 |
383,614 |
57,842 |
441,456 |
2016 |
227,144,466 |
27,006,670 |
505,498,274 |
422,384 |
63,919 |
486,303 |
2017 |
229,254,924 |
26,398,281 |
509,804,341 |
392,148 |
57,868 |
450,016 |
2018 |
230,694,063 |
27,379,324 |
513,726,592 |
381,382 |
56,054 |
437,436 |
2019 |
226,740,688 |
25,689,603 |
510,503,678 |
365,038 |
53,230 |
418,268 |
Totals |
3,610,450,803 |
465,008,694 |
7,898,488,703 |
6,229,280 |
952,221 |
7,181,501 |
- The authors also stated “ Complex weights were applied to estimate the entire population, resulting in fractional values”. – Yet it is not clear how “complex weight were applied”. There should be clear explanation regarding how the 2003-2019 survey data were combined, weighted percentages was calculated what for what variables and which groups
Author response: Thank you for your valuable recommendation. We previously referenced the reader to the BRFSS site for this information but have added the following statements to address your concerns. We also show how the weights are used to provide population estimates in Table 1. We are hopeful that these actions have addressed your concerns.
Complex weights based on location and socio-demographic information were applied to estimate the entire population, resulting in fractional values. Prior to 2011, samples were weighted by stratum (area code / prefix combinations) times the inverse number of telephones in the household times the number of adults in the household times the number of people in an age-by-sex or age-by-race/ethnicity-by-sex category in the population of a region or a state divided by the sum of the preceding weights for the respondents in the same age-by-sex or age-by-race/ethnicity-by-sex category (adjusting for noncoverage and nonresponse and forces the sum of the weighted frequencies to equal population estimates for the region or state.) Two separate variable weight fields. This weighting was applied through the use of separate fields for stratum and weights. From 2011 onwards, raking replaced the post-stratification process. Raking allows for incorporation of cellular telephone survey data and allows for inclusion of other demographic considerations to improve so-cio-demographic matching by stratum.
Reviewer comment 7: The independent variable: It is stated: “The primary variables of interest were the year of the estimate (2003 through 2018), and Veteran status, coded {0 = non-Veteran, 1= Veteran} with blank, refused, and do not know coded as non-Veteran”
- Currently it isn’t clear how veteran status variable was obtained for this study?
Author response: Thank you for your obvservation and question. You certainly bring up a good point and as a result, our study team now has both a table of variables, associated definitions, and BRFSS question.
The primary variables of interest were the year of the estimate (2003 through 2019), and Veteran status, recoded {0 = did not self-report as veteran, 1= self-reported as veteran}. This coding results from the survey questions that follow.
- Years 2003 through 2006 (Code: VETERAN). Have you ever served on active duty in the United States Armed Forces, either in the regular military or in a National Guard or military reserve unit?” 1=Yes, 2=No, 7=Don’t Know / Not Sure, 9=Refused, Blank=Not asked / Missing
- Years 2007 through 2008 (Code: VETERAN1). “Have you ever served on active duty in the United States Armed Forces, either in the regular military or in a National Guard or military reserve unit? Active duty does not include training for the Reserves or National Guard, but DOES include activation, for example, for the Persian Gulf War.” 1=Yes, 2=No, 7=Don’t Know / Not Sure, 9=Refused, Blank=Not asked / Missing
- Years 2009 (Code VETERAN2). “Have you ever served on active duty in the United States Armed Forces, either in the regular military or in a National Guard or military reserve unit? Active duty does not include training for the Reserves or National Guard, but DOES include activation, for example, for the Persian Gulf War.” 1=Yes, now on Active Duty, 2=Yes, on Active Duty during the last 12 months but not now, 3=Yes, on active duty in the past, but not during the last 12 months, 4=No, training for Reserves or National Guard only, 5=No, never served in the military, 7=Don’t Know / Not Sure, 9=Refused, Blank=Not asked / Missing
- Years 2010 through 2019 (Code: VETERAN3). “Have you ever served on active duty in the United States Armed Forces, either in the regular military or in a National Guard or military reserve unit? Active duty does not include training for the Reserves or National Guard, but DOES include activation, for example, for the Persian Gulf War.” 1=Yes, 2=No, 7=Don’t Know / Not Sure, 9=Refused, Blank=Not asked / Missing
In all years, the responses were identically coded with the exception of 2009. Recoding for this year assigned any yes values equal to 1 and all other values equal to zero. Due to the exceedingly low numbers of responses that were blank, refused, and “do not know” (e.g., 0.574% in 2019), these were imputed with the modal response of non-Veteran (86.7% of the population in 2019).
- Based on what question/ information from the BRSFF data base was it constructed?
Author response: Please see above. We also now include a new Table to define all variables for the base year definition for coding purposes. Thank you so much for your question.
- Regarding the re-coding “blank, refused, and do not know coded as non-Veteran” Blank are typically state that were not asked the questions, refused and don’t know are consider missing data. First, it is recommended to report the numbers; in addition typically those responses are excluded from the analysis. It is not clear way it was re-coded as non-Veteran. Could the authors justify the decision?
Author response: This is a great question and one that we certainly should have addressed in the manuscript. The proportion in these categories are negligible (a fraction of a percent) and might have been omitted; however, we preferred to impute them as the modal category (which is probabilistically non-veteran). Based on your question, our Study Team has added the following.
Due to the exceedingly low numbers of responses that were blank, refused, and “do not know” (e.g., 0.574% in 2019), these were imputed with the modal response of non-Veteran (86.7% of the population in 2019).
We also now demonstrate the unknown / refused missing fractions in the paper. Only in the case of overweight / obese reporting are the values greater than 1%. Thus we coded 1=positively identified as having X versus 0=did not positively identify as having X.
Table 3. Year 2019 dependent variable proportions are shown (sample / population estimate).
Question * |
Yes |
No |
Unknown |
Overweight / Obese |
62%/60% |
29%/30% |
9%/10% |
Coronary Heart Disease |
6%/4% |
93%/95% |
1%/1% |
Stroke |
4%/3% |
95%/96% |
1%/1% |
Skin Cancer |
10%/6% |
90%/93% |
0%/0% |
Other Cancer |
10%/7% |
90%/93% |
0%/0% |
COPD |
8%/7% |
91%/93% |
1%/0% |
Arthritis |
33%/25% |
66%/75% |
1%/0% |
Mental Health |
34%/37% |
66%/63% |
0%/0% |
Kidney Issues |
4%/3% |
96%/97% |
0%/0% |
Diabetes |
14%/11% |
86%/89% |
0%/0% |
Reviewer comment 8: Dependent variables: It is stated: “Dependent variables in the study were dichotomously coded (0 = absence of morbidity, 1 = presence of morbidity) and included overweight / obese, heart disease, stroke, skin cancer, cancer, COPD, arthritis, mental health condition, kidney disease, and diabetes. In all cases, “do not know”, “refused”, “unknown”, and “missing” responses were coded 0 (not present).”
- Currently it isn’t clear how morbidity status was obtained for this study?
Author response: Our Study Team apologizes for not being clear in this respect. “Morbidity” was a generic term for any of the possible selected dependent variables (Table 3). It was not a variable in itself. To be sure there was no confusion, we added the following.
All dependent variables in the study were dichotomously coded. In all cases, the categories of “Don't Know/Not Sure” and “Not Asked / Missing” were coded with the modal response, as the proportion of these values was 1% or less. For obesity, the modal response was greater than 25% body mass index. For all other dependent variables, the modal response was “No.” For all variables other than obesity, the proportion missing was 1% or less (negligible). For obesity, about 10% of the observations were in the cat-egories “Don't Know/Refused/Missing.” The modal response reflecting overweight / obesity status (greater than 60% of the respondents) is likely to best categorize these in-dividuals. Table 3 shows the unweighted and weighted proportions in the categories of “Yes”, “No”, and “Unknown” for the dependent variables in year 2019.
- Based on what questions/ information from the BRSFF database was it constructed?
Author response: Thank you for pointing this. In response to this, we added Table 2 which should clarify this.
- Regarding the re-coding ““do not know”, “refused”, “unknown”, and “missing” responses were coded 0 (not present)” – First it is recommended to report the numbers; in addition typically those responses are excluded from the analysis. It is not clear way it was re-coded as non- present. Could the authors justify the decision?
Author response: Thank you and yes. Please see Table 3 for these numbers in the Year 2019. Because they are 1% or less for all variables in the study except overweight / obesity, we imputed the modal responses. For overweight / obesity, there were <10% in these categories, so we assigned them the modal response (overweight / obese) as well. We hope this clarifies any confusion in this matter.
Table 3. Year 2019 dependent variable proportions are shown (sample / population estimate).
Question * |
Yes |
No |
Unknown |
Overweight / Obese |
62%/60% |
29%/30% |
9%/10% |
Coronary Heart Disease |
6%/4% |
93%/95% |
1%/1% |
Stroke |
4%/3% |
95%/96% |
1%/1% |
Skin Cancer |
10%/6% |
90%/93% |
0%/0% |
Other Cancer |
10%/7% |
90%/93% |
0%/0% |
COPD |
8%/7% |
91%/93% |
1%/0% |
Arthritis |
33%/25% |
66%/75% |
1%/0% |
Mental Health |
34%/37% |
66%/63% |
0%/0% |
Kidney Issues |
4%/3% |
96%/97% |
0%/0% |
Diabetes |
14%/11% |
86%/89% |
0%/0% |
*May not add to 100% due to rounding
- The authors’ state: “Many of the dependent variables did not have observations dating back to 2003. Table 2 shows the analysis starting year for each of the variables.” - However could the authors please clarify did the information was collected across all 50 states as well as the District of Columbia and 3 U.S. territories
Author response: Thank you for your question. Our Study Team used complete data and now specify that these include the District of Columbia (DC) and the three U.S. territories.
- The authors combined data analysis and other potential variables under dependent Which is extremely confusing and hard to follow. It would be recommended that the authors would create two separate subheading. For example first subheading: Study Covariates for example: here included race/ethnicity (white, black and other race/ethnicity), age (18–24, 25–34, 35–44, 45–54, 55–64) etc... In addition, it would recommend excluding the SASS variable code for example “_IMPRACE”. The second subheading: Data Analysis- that include clear description of data analysis that been carried in the study. That isn’t clear.
Author response: Thank you so much for this observation. In an effort to better clarify this area, our Study Team has inserted separate headings as follows:
2.1. Data Source
2.2. Study Sample
2.3. Study Measures
2.3.1. Dependent Variables
2.3.2. Independent Variables
2.4. Methods and Models
2.4.1. Time Series
2.4.2. General Linear Models
2.4.3. Recoding for Covariates
2.5. Data Analysis
Reviewer comment 9: Results
It isn’t clear -How much participants included in the study; how much was in the veterans ? how much was in the non-veterans , how much was in each chronic conditions. The authors should consider present table of weighted percentages for demographics and chronic conditions variables by veteran and non-veteran
Author response: Thank you for your observation. In order to address your concern, our Study Team revamped Table 1 to illustrate population estimates and sample sizes by Veteran status. We are hopeful that this addressed your concerns and made it more understandable to the Reader.
Year |
Non-Vet Population |
Vet Population |
Population Estimate |
Non-Vet Sample |
Vet Sample |
Sample |
2003 |
190,348,049 |
30,003,072 |
220,351,121 |
228,159 |
36,525 |
264,684 |
2004 |
191,637,278 |
29,746,086 |
441,734,485 |
260,982 |
42,840 |
303,822 |
2005 |
194,578,583 |
29,532,523 |
445,494,470 |
305,107 |
51,005 |
356,112 |
2006 |
198,138,945 |
29,118,914 |
451,368,965 |
304,989 |
50,721 |
355,710 |
2007 |
202,498,717 |
27,673,461 |
457,430,037 |
370,990 |
59,922 |
430,912 |
2008 |
205,615,985 |
27,244,684 |
463,032,847 |
358,433 |
56,076 |
414,509 |
2009 |
208,756,506 |
26,249,349 |
467,866,524 |
374,909 |
57,698 |
432,607 |
2010 |
211,037,577 |
26,048,662 |
472,092,094 |
390,643 |
60,432 |
451,075 |
2011 |
212,198,501 |
25,812,791 |
475,097,531 |
441,873 |
64,594 |
506,467 |
2012 |
216,959,427 |
26,098,283 |
481,069,002 |
415,817 |
59,870 |
475,687 |
2013 |
219,968,409 |
26,056,006 |
489,082,125 |
430,268 |
61,505 |
491,773 |
2014 |
220,704,167 |
27,778,365 |
494,506,947 |
402,544 |
62,120 |
464,664 |
2015 |
224,174,518 |
27,172,620 |
499,829,670 |
383,614 |
57,842 |
441,456 |
2016 |
227,144,466 |
27,006,670 |
505,498,274 |
422,384 |
63,919 |
486,303 |
2017 |
229,254,924 |
26,398,281 |
509,804,341 |
392,148 |
57,868 |
450,016 |
2018 |
230,694,063 |
27,379,324 |
513,726,592 |
381,382 |
56,054 |
437,436 |
2019 |
226,740,688 |
25,689,603 |
510,503,678 |
365,038 |
53,230 |
418,268 |
Totals |
3,610,450,803 |
465,008,694 |
7,898,488,703 |
6,229,280 |
952,221 |
7,181,501 |
Reviewer comment 10: Discussion
The author’s state: “Multiple deployments over the last several decades may be a major contributor to increased illness seen after separation from the service [30]. The demand placed upon the U.S. Service member, particularly since the mid to late 1980’s has been illustrated in the number of military operations requiring deployment [20]. Unfortunately, the number of available personnel to support these deployments has not kept pace, resulting in more Service members deploying more often. As depicted in the health outcomes data, this combination of increased demand in military operations placed upon a smaller population may be resulting in a group of at-risk individuals for a variety of debilitating health conditions [1]. Additional stressors may exist in the occupational requirement for all Service members to meet stringent height and weight standards, and physical fitness standards, both of which are evaluated every six months [17]. Perhaps the combination of each of these stressors ultimately contribute to the stark differences in health outcomes between the Veteran populations compared to non-Veteran populations [1-3] As depicted in the health outcomes data, this combination of increased demand in military operations placed upon a smaller population may be resulting in a group of at-risk individuals for a variety of debilitating health conditions [1]. Additional stressors may exist in the occupational requirement for all Service members to meet stringent height and weight standards, and physical fitness standards, both of which are evaluated every six months [17].”
Yet the study didn’t examine any of the variables [Information on deployment or height and weight standards, and physical fitness standards] the authors cannot provide data. This section should be remove.
Author response: Thank you for your comments and although the Study Team fully appreciates the Reviewer’s suggestion of removal of this section, it is our collective belief that the revamped tables and analyses strengthen the argument the Service members throughout this period have faced an increasing number of stressors which include maintaining a high level of physical fitness, maintaining strict height and weight standards, and an increase in operational deployments which inherently include potential family separation. Removal of this section would greatly weaken this argument.
Reviewer 2 Report
The collective health of those who served in the U.S. military exhibits a number of health conditions that paint a concerning picture of morbidity among the U.S. Veteran population. The combination of increased demand in military operations placed upon a smaller population may be resulting in a group of at-risk individuals for a variety of debilitating health conditions. So the aim of the study was to explore health outcomes for U.S. Veterans compared to Non-Veterans from 2003 to 2019.
The paper has a clear friendly structure (Introduction, Materials and Methods, Results, Discussion and Conclusions). The subject is interesting and useful and raises the worrying issue of the collective poor health of the U.S. Veterans. The introduction is quite informative and the subject is thoroughly explained. The manuscript stands good in- depth analysis of the results and transparent and sufficient discussion containing the limitation section. The text is complemented by three tables, six figures and enriched with 34 adequate references.
However there are a few issues that could be supplemented:
- Materials and Methods section: could you please explain on what basis did you decide:
- as regards ‘marital status’ - the small proportion of “refused to answer” or left blank were assigned to the modal category (unweighted), 1 = married ?;
- as regards ‘education status’ - the small proportion of “refused to answer” or left blank were assigned to the modal category (unweighted), 6 =college 4 or more years?;
- as regards ‘income’ the small proportion of refusals and ‘don`t know’ were assigned to the modal category (unweighted), 1= employed for wages?
- Discussion: In my opinion it would be useful to focus more deeply on the important question: why rates of self-reported mental health disorders were not higher among Veterans than among non-Veterans, since it is well known that military service can often result in posttraumatic stress disorders, e.g. depression and other mental health disorders among veterans?
Author Response
Thank you for giving us the opportunity to submit a revised draft of the manuscript, “Exploring Health Outcomes for U.S. Veterans Compared to Non-Veterans from 2003 to 2019“ for publication in Healthcare. We appreciate the time and effort that you and your reviewers dedicated to providing feedback on our manuscript and are grateful for the insightful comments on the valuable improvements to our paper. We have incorporated the recommendations made by the reviewers. These changes are highlighted within the manuscript for your ease in reading. Please see below (in blue text), for a point-by-point response to the reviewers’ comments and concerns. All page numbers refer to the revised manuscript file with tracked changes.
Reviewer comment 1: Materials and Methods section: could you please explain on what basis did you decide:
- as regards ‘marital status’ - the small proportion of “refused to answer” or left blank were assigned to the modal category (unweighted), 1 = married ?;
- as regards ‘education status’ - the small proportion of “refused to answer” or left blank were assigned to the modal category (unweighted), 6 =college 4 or more years?;
- as regards ‘income’ the small proportion of refusals and ‘don`t know’ were assigned to the modal category (unweighted), 1= employed for wages?
Author response: We are very grateful for your question. Out of 418,268 survey observations in 2019, less than 1% of respondents did not respond / refused to respond to marital status, less than 1% did not respond / refused educational level, and less than 10% did not respond to income questions. While imputing the mode makes absolute sense for central tendency, we now refer to these questions as 1=self-identified as married, 0 otherwise; 1=self-identified as a college graduate, 0 otherwise, and 1=self-identified in the modal category as having earned $75K+, 0 otherwise. The re-coding avoids imputation as it is the event A and its associated complement, Ac.
Reviewer comment 2: Discussion: In my opinion it would be useful to focus more deeply on the important question: why rates of self-reported mental health disorders were not higher among Veterans than among non-Veterans, since it is well known that military service can often result in posttraumatic stress disorders, e.g. depression and other mental health disorders among veterans?
Author response: Thank you for your valuable recommendation. Of course, what you speculate is true. More often than not, many veterans believe that they are mentally healthy when in reality they may be suffering from some of the conditions you state (PTSD, depression). Our Study Team addressed this when we stated in the discussion that “It is possible that the negative consequences of reporting mental health issues in the military (e.g., security clearance revocation) continue to play a role in how persons reported in this dataset.” However, your point is well-taken and we will recommend this area as an important area for future study.
Reviewer 3 Report
Important study identifying relevant stressors for veterans and the implicated longer term health challenges. Interesting finding about lower mental health reported; this is hard to believe and could be expanded upon in the Discussion section. Methods to evaluate data seem effective although hard to follow in some parts of presentation. Clear strengths and limitations and could expand recommendations for future research.
Author Response
Thank you for giving us the opportunity to submit a revised draft of the manuscript, “Exploring Health Outcomes for U.S. Veterans Compared to Non-Veterans from 2003 to 2019“ for publication in Healthcare. We appreciate the time and effort that you and your reviewers dedicated to providing feedback on our manuscript and are grateful for the insightful comments on the valuable improvements to our paper. We have incorporated the recommendations made by the reviewers. These changes are highlighted within the manuscript for your ease in reading. Please see below (in blue text), for a point-by-point response to the reviewers’ comments and concerns. All page numbers refer to the revised manuscript file with tracked changes.
Reviewer comment. Important study identifying relevant stressors for veterans and the implicated longer term health challenges. Interesting finding about lower mental health reported; this is hard to believe and could be expanded upon in the Discussion section. Methods to evaluate data seem effective although hard to follow in some parts of presentation. Clear strengths and limitations and could expand recommendations for future research.
Author response: Thank you for your insightful comment and observation. It is our Study Team’s collective belief that the revamped tables and analyses strengthen the argument the Service members throughout this period have faced an increasing number of stressors, resulting in many, poor physical health outcomes: the BRFSS data illustrate this. However, it is intriguing that the BRFSS data do NOT reflect poor MENTAL health outcomes in the BRFSS data set. Again, our Study Team suspects that this may be a result of the reluctance of Service members to report mental health conditions for fear of negative consequences such as loss of security clearance or loss of occupation. We have recommended further research in the future be focused on this area of mental health and the Service member.